# What drives the effectiveness of social distancing in combating COVID-19 across U.S. states?

**Mu-Jeung Yang** [1]*, **Maclean Gaulin** [2], **Nathan Seegert** [2], **Yang Fan** [3]

**1** Department of Economics, University of Oklahoma, Norman, Oklahoma, United States of America,
**2** David Eccles School of Business, University of Utah, Salt Lake City, Utah, United States of America,
**3** Colby College, Waterville, Maine, United States of America

* mjyang@ou.edu

## Abstract

We propose a new theory of information-based voluntary social distancing in which people's responses to disease prevalence depend on the credibility of reported cases and fatalities and vary locally. We embed this theory into a new pandemic prediction and policy analysis framework that blends compartmental epidemiological/economic models with Machine Learning. We find that lockdown effectiveness varies widely across US States during the early phases of the COVID-19 pandemic. We find that voluntary social distancing is higher in more informed states, and increasing information could have substantially changed social distancing and fatalities.

## Introduction

As the recent COVID-19 pandemic shows, non-pharmaceutical interventions (NPIs) such as officially mandated social distancing policies (henceforth "lockdowns") can play a critical role for saving lives, especially in the early stages of a pandemic, during which an effective vaccine is not yet available. In this context, policy makers face an important trade-off between lives and livelihoods, as lockdowns will also reduce mobility and related local economic activity. This paper seeks to provide a modelling framework to quantify the trade-off between health benefits and economic costs of lockdown policies, by combining a theory of information-based voluntary social distancing within a compartmental model epidemiological model with a Machine Learning approach to prediction.

Broadly, there are two approaches to modeling and forecasting pandemic dynamics to inform policy, as evidenced by predictions in the wake of COVID-19.[1] First, purely "statistical models" seek to fit a flexible functional form to data such as daily deaths. Prominent examples include the "curve fitting" approach by the Institute of Health Metrics and Evaluation at the University of Washington (IHME, see [2]) or by White House economist Kevin Hassett (see [3]). Purely statistical approaches are adaptable but also tend to be strongly driven by functional form assumptions that make these approaches prone to overfitting. Both the approaches by the IHME and by Hassett strongly underpredicted COVID-19 fatalities with the pandemic effectively ending before June 2020 in the USA. At the same time,

more sophisticated Machine Learning approaches can be used to, reduce overfitting. A more fundamental challenge, even to pure Machine Learning prediction models, is that these approaches are "black box" models in nature and therefore neither allow an understanding of how and why policy interventions will work nor enable cost-benefit analyses of lockdown policies.

**Data availability statement:** The data for this paper is available at https://github.com/gaulinmp/covid_state_heterogeneity

**Funding:** This work was supported by the Centers for Disease Control and Prevention (CDC) (Centers for Outbreak Analytics and Disease Modeling). Grant: 1 NU38FT000009-01-00 (Awarded to NS). Financial support for publication was provided by the University of Oklahoma Libraries' Open Access Fund.

Second, traditional "compartmental models" used in epidemiology ([4–6]) have biological foundations and can allow researchers to better understand how and why lockdown policies work. However, early off-the shelf compartmental models of COVID-19 strongly overpredicted infections and forecasted that up to 70% of the US population will be infected within months in the absence of lockdowns ([2]). This overprediction is a direct consequence of the exponential growth of epidemics built into the basic structure of the simplest compartmental models. A key feature missing from these early compartmental models is voluntary social distancing, defined as the tendency of people to reduce mobility to avoid infection even in the absence of a lockdown ([7]). [8] have shown that voluntary social distancing can not only explain the "sub-exponential" growth of COVID-19 in the data, but also leads to lower estimates of the causal impact of lockdown policies on COVID-19 fatalities. This latter result is driven by the fact that even without lockdowns, voluntary social distancing reduces the spread of the virus. However, the approach of [8] leaves several questions open. On the one hand, [8] models individual mobility decisions in an "ad hoc" or "behavioral" manner, so that it remains unclear whether and how policy can influence voluntary social distancing. On the other hand, [8] focus on a single location (using national data on the UK) so that they do not analyze how the effectiveness of lockdown policies varies with local characteristics, such as local differences in voluntary social distancing.

This paper complements the existing literature through three distinct contributions. Our first contribution is a new theoretical model of information-based voluntary social distancing, in which people learn about infection risks from publicly available data on new cases and fatalities. Importantly, the strength of voluntary social distancing responses to reported local cases and fatalities is driven by the perceived credibility of this data, which can be influenced by local governments' information policies and which will vary locally. In this context, our model predicts that local information policies that produce less credible case and fatality counts lead to a lower signal-to-noise ratio of published case and fatality counts. As a consequence, local information policies will impact the degree of voluntary social distancing directly and the effectiveness of lockdown policies indirectly. This insight goes well beyond the analysis of [8] and has important policy implications. We are not aware of any other paper in the literature providing this insight or any similar policy analysis.[2]

Our second contribution is methodological, in that we propose a new prediction and policy analysis framework combining an extended compartmental epidemiological model with Machine Learning. We embed our theory of information-based social distancing into a rich compartmental model including a variety of often unobserved, time-varying factors such as asymptomatic transmission, symptom-based testing and quarantining, and time-variation in fatality rates. In addition to daily local data on new COVID-19 cases and fatalities, we calibrate the model to local, daily cellphone-based data on mobility, which we establish as valid proxy for economic activity. Our framework can therefore provide cost-benefit analysis for policy makers interested in the economic and health effects lockdown policies. To increase the reliability of model forecasts, we use ensemble learning and cross-validation, two ideas from Machine Learning. Specifically, we re-calibrate the model for different time horizons and then use a weighted average of these estimates, to increase the robustness of out-of-sample predictions. The ensemble weights are then chosen via cross-validation, which

minimizes the out-of-sample prediction error. Using these additional steps allows us to generate more generalizable patterns that are less likely driven by statistically noisy initial conditions. Of the 136 COVID-19 forecasting models reviewed in [1], only 3 papers use a hybrid approach between Machine Learning and compartmental models in combination with mobility data. All of these papers use mobility data as a predictor for new cases and fatalities but none of these three papers allows for a feedback of case and fatality data to mobility, the way our paper does. This feedback is a direct consequence of our theory of information-based voluntary social distancing and is therefore a main focus of our analysis.

Our third contribution consists of novel empirical results from the application of our new model and methodology to the 50 US states during the early phase of COVID-19 in 2020. We find that local differences in mobility responses and other parameters imply a wide variation of the economic efficiency of lockdown policies across states. To quantify economic efficiency of lockdown policy (henceforth "lockdown efficiency") we use mobility as a proxy for economic activity and measure mobility lost if lockdowns would have been used to save the same number of lives as have been saved through information-based voluntary social distancing in a given state. We find that lockdown efficiency varies widely. For example, for US states in the 25th percentile of the lockdown efficiency distribution, lockdowns could have saved the same number of lives as voluntary social distancing, but allowing for 75% more mobility until June 2020. In contrast, for US states in the 75th percentile of the lockdown efficiency distribution, using lockdowns instead of voluntary social distancing would have implied almost 20% less economic activity. In other words, the efficiency of lockdown policies varies substantially across states. To investigate the role of differences in local information-based voluntary social distancing, we impose different mobility responses to reported case and fatality counts. Our information-based voluntary social distancing model suggests that these mobility responses can be systematically influenced by the credibility of local government reports. To provide a reasonable counterfactual for changes in credibility of local government reports, we compare the number of lives saved by uniformly imposing voluntary social distancing responses in places with low information transmission like West Virginia compared to relatively high information transmission places like Massachusetts. Using West Virginia parameters for the whole US implies over 240,000 additional fatalities before June 2020. In contrast, imposing parameters from Massachusetts across all US states only saves an additional 24,000 lives before June 2020. This shows that the existence of important cost-benefit asymmetries of these information policies, as low-credibility information policies are much more costly in terms of fatalities than high-credibility information policies are beneficial in saving lives. Our quantitative evidence on how local information policies effect pandemic outcomes, to our knowledge unique to this study, highlights the importance of credible information due to voluntary social distancing.

## 1 Motivating stylized facts

On March 11, 2020, the World Health Organization (WHO) declared COVID-19 a global pandemic. California was the first US state to order a state-wide lockdown on March 19, 2020, with nearly all other states following in the next 2–3 weeks. These lockdown orders varied from mandatory (e.g., California) to voluntary (e.g., Utah) to none at all (e.g., Arkansas), with significant heterogeneity in the application and severity of the orders at various levels of government (for example, Utah did not have a mandatory lockdown, but Salt Lake City did). Irrespective of lockdown enforcement, households might infer from lockdowns a signal about the severity of the outbreak, which is why we think of lockdown-induced social distancing.

Thus the extent to which citizens followed mandatory or voluntary social distancing safety measures is ultimately an empirical question.

To quantitatively measure people's response, we capture the extent of social distancing by using cellphone-location based mobility data from Google.[3] The Google mobility measures provide a daily-frequency comparison of mobility relative to the same calendar day in 2019, to control for general seasonal patterns. Google provides this mobility data for different geographic locations and different categories of points of interest. We focus on economically relevant categories, such as mobility for work, grocery shopping, retail shopping (including restaurants), and transportation (such as public transit). We exclude categories such as "parks", since outdoor disease transmission is less common and mobility within parks has increased in some states during COVID-19.[4]

Our first stylized fact is that social distancing quantitatively matters, but lockdown-induced social distancing cannot fully explain it. This fact suggests that individuals' behavior needs to be explicitly modeled to capture social distancing and the spread of COVID-19 properly. Fig 1 provides an event-study graph of mobility for economic activities listed above (henceforth "mobility") for all 50 US states. The vertical red line centers the graph around

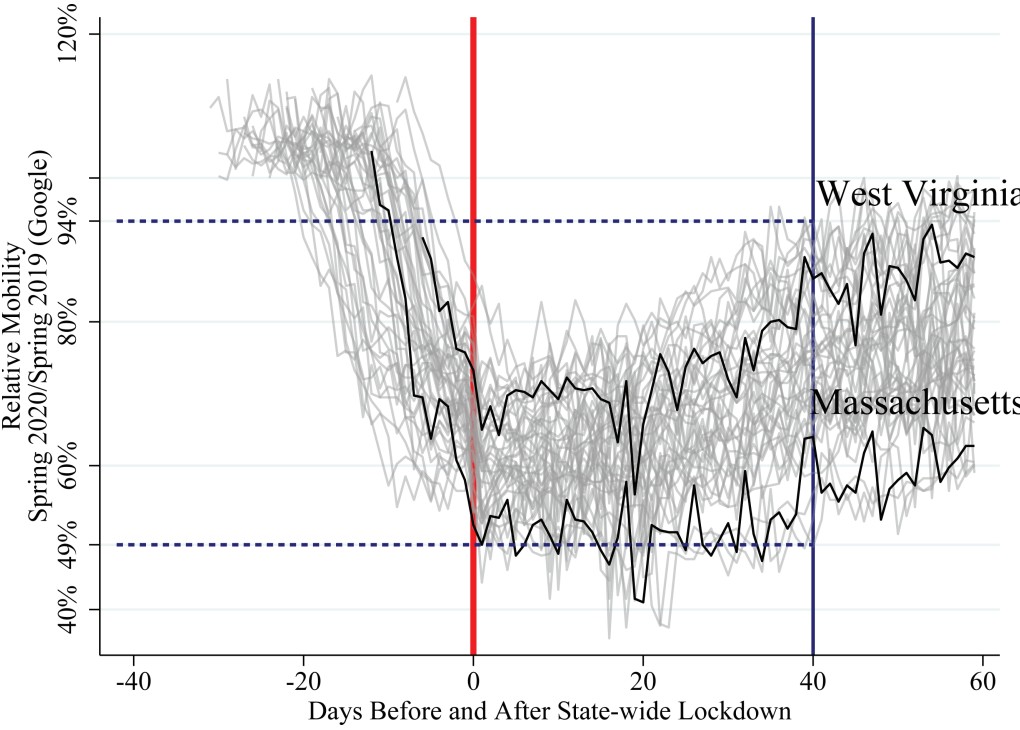

**Fig 1. Voluntary social distancing before effective date of state-wide lockdown.** *Note:* This figure uses cellphone-location based mobility data from Google to quantitatively measure people's response (see: https://www.google.com/covid19/mobility/). The Google mobility measures provide a daily-frequency comparison of mobility relative to the same calendar day in 2019, to control for general seasonal patterns. A value of 70% is interpreted as mobility on this day in 2020 is 70% the mobility on this day in 2019. We focus on economically relevant categories, such as mobility for work, grocery shopping, retail shopping (including restaurants), and transportation (such as public transit) and exclude categories such as "parks," since outdoor disease transmission is less common. The mobility data is centered around the day a state-wide lockdown is imposed (given by the bold red line). To demonstrate heterogeneity across states, we denote the difference in mobility 40 days after a state-wide lockdown (thin blue line, with dashed horizontal lines that denote the range). The mobility data for two states, Massachusetts and West Virginia, are given in black and labeled.

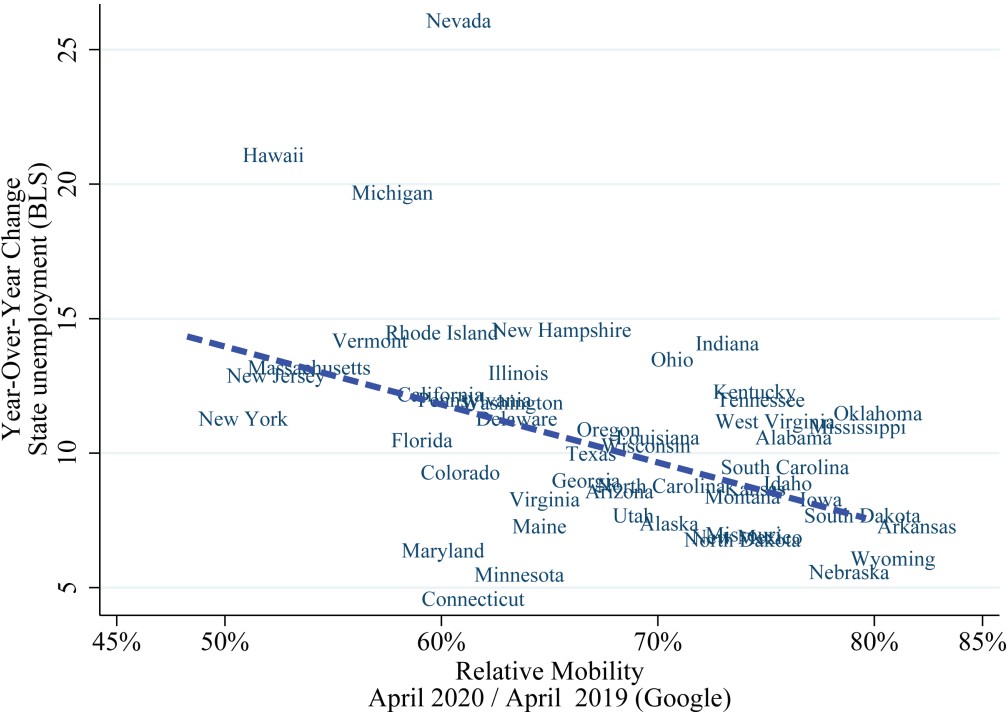

**Fig 2. Mobility and unemployment rates across states (April, 2020).** *Note:* This figure uses cellphone-location based mobility data from Google to quantitatively measure people's response (see: notes for Fig 1). State unemployment data comes from the Bureau of Labor Statistics (BLS).

the day each state imposed its lockdown, and the vertical-axis measures mobility relative to 2019. Each grey line is the daily relative mobility for a different US state, with West Virginia and Massachusetts in black for comparison. Fig 1 shows that mobility has substantially fallen in all 50 states. On average, mobility drops from above 100%, 20 days before the lockdowns, to a nadir of 60% and gradual increase to 70%, 40 days after a state lockdown. Much of the fall in mobility, however, pre-dates the imposition of official state lockdowns. Taken together, Fig 1 suggests mobility changed dramatically and is influenced by factors other than official lockdown policy, see [10].

Our second stylized fact is that changes in mobility vary greatly across states to an economically relevant extent. While all states follow a similar pattern, the differences between state responses are noticeable in Fig 1, where the spread is 45% between states 40 days after a state-wide lockdown. There are many potential reasons for this heterogeneous response. Two composite reasons include the characteristics of the local outbreak, such as the number of confirmed cases and population density, and the beliefs of the state residents, which are influenced by information from federal and state officials as well as different news sources. These heterogeneous mobility responses, in turn, matter for state-level unemployment, as Fig 2 shows. Mobility is, therefore, a useful proxy for how social distancing affected economic activities.

Our third stylized fact is that the differences in mobility between states correlate strongly with their ex-ante beliefs. To investigate differences in mobility due to differences in beliefs, we exploit the current political climate as an observable signal of the beliefs about the virus. For example, it is plausible that areas with a higher approval rating for President Trump may

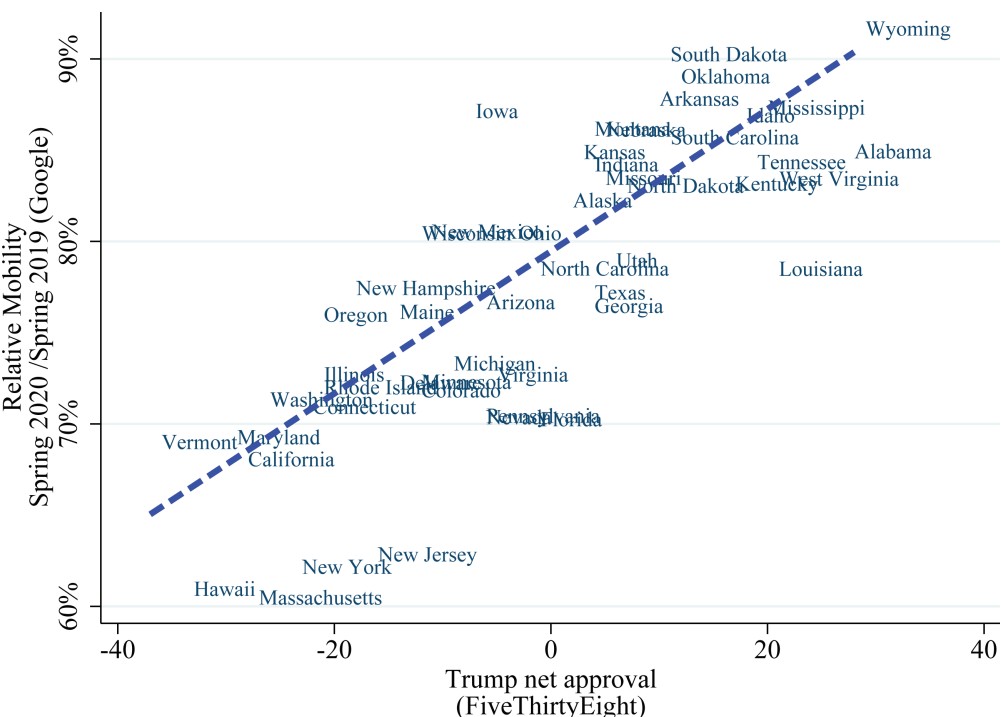

**Fig 3. Mobility and President's approval rating across states.** *Note:* This figure uses cellphone-location based mobility data from Google to quantitatively measure people's response (see: notes for Fig 1) and approval ratings for the President from FiveThirtyEight (https://github.com/fivethirtyeight/data/tree/master/trump-approval-ratings) averaged over the spring of 2020.

have a different belief about the virus than other areas because of the messages the President has given.[5] We therefore use net presidential approval ratings for President Trump in April as a measure of locally perceived credibility of the COVID-19 threat. This correlation is shown in Fig 3. The horizontal axis is the President's net approval rating, and the vertical axis is the average relative mobility until June 2020.

Fig 3, however, cannot tell us whether these differences are driven by differences in voluntary social distancing or the strength of local lockdown measures. To provide simple reduced-form evidence on this question, we run the following regression, separately for each state $s$:

$$m_{s,t} = \mu_{s,0} + \mu_{s,1} \cdot \ln O_{s,t} + \mu_{s,2} \cdot \ln F_{s,t} - \lambda_t + e_{s,t}, \tag{1}$$

where $m_{s,t}$ is mobility, $\ln O_{s,t}$ is the log of confirmed case counts, $\ln F_{s,t}$ the log of cumulative fatalities, $\lambda_t$ the coefficient estimated on a time dummy that is one during the duration of state-wide lockdowns and $e_{s,t}$ is an error term. This reduced form model provides a first pass at quantifying differences in voluntary social distancing across states in response to public information on local case and fatality counts while controlling for state-wide lockdown measures.[6] It should be noted that despite the simplifying assumptions of this reduced form model of mobility, the median $R^2$ is around 71%. Such a high in-sample $R^2$ lends credence to this simple model, which explains the vast majority of mobility variation. When we exclude the lockdown policy dummies, this average $R^2$ falls from 71% to 46%, implying that

lockdowns and voluntary social distancing seem to be jointly important in understanding mobility responses.

We visualize equation (1) in Figs 4 and 5 with log of confirmed cases on the horizontal axis and mobility on the vertical axis. We display the variation across time within Massachusetts and West Virginia in Fig 4. Both states experience lower mobility as log-confirmed cases increase, but Massachusetts is more responsive (given by a steeper slope $\mu_1$) and has a higher mobility in the absence of log-confirmed cases (given by higher vertical-intercept $\mu_0$). We display variation across states in Fig 5, where we use each state's average log confirmed cases and mobility. We also add a linear fit trend line which represents the average across states, and has negative slope suggesting that the negative relation holds both across states and within states over time. The trend line provides the average responsiveness and states above the trend line are less responsive than average (e.g., Wyoming and West Virginia) and states below the trend line are more responsive (e.g., Massachusetts and Vermont).

Equation (1) naturally separates out the initial mobility response ($\mu_{s,0}$) and the mobility responses to published local confirmed case counts and fatalities ($\mu_{s,1}, \mu_{s,2}$). It should be noted that $\mu_{s,1}$ captures what [7] calls the "prevalence elasticity," which is the response of risky behavior leading to exposure, to disease prevalence. We define "responsiveness" to case counts as the absolute value of the prevalence elasticity, or $|\mu_1|$. Figs 6–8 show that responsiveness to case counts systematically differs across states. The responsiveness decreases with the President's net approval rating and increases with education attainment. Further, and perhaps surprisingly, responsiveness $|\mu_1|$ is positively correlated with initial mobility, Fig 8.

A potential reason that states end up with high responsiveness and high initial mobility is that people who trust the reported cases are able to have both high mobility when cases are

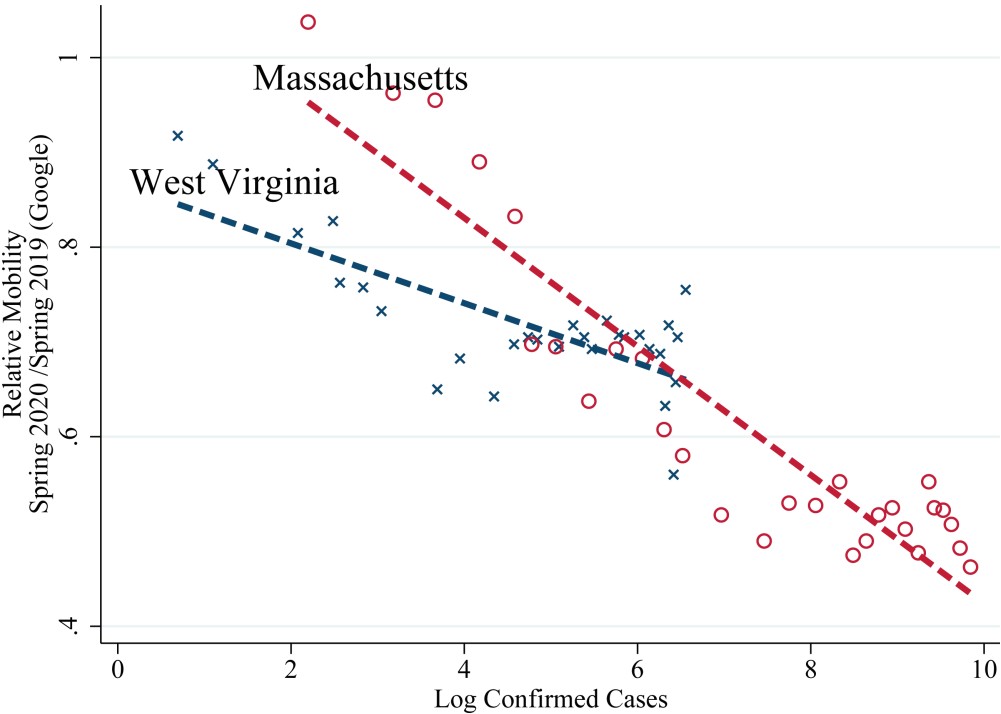

**Fig 4. Mobility and log confirmed cases across time in MA and WV**. *Note:* This figure uses cellphone-location based mobility data from Google to quantitatively measure people's response (see: notes for Fig 1).

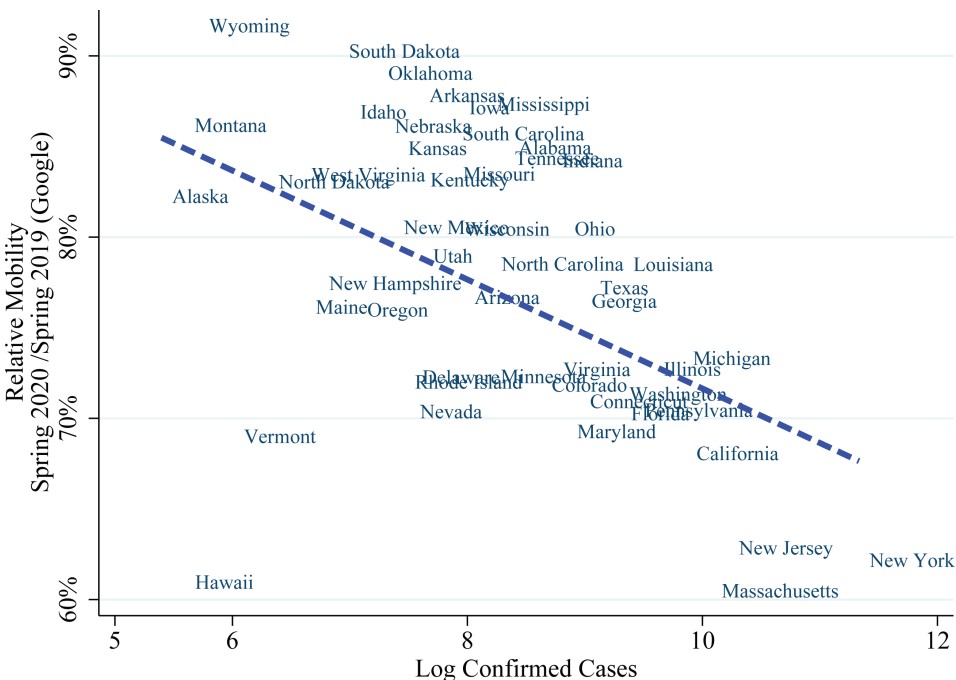

**Fig 5. Mobility and log confirmed cases across states.** *Note:* This figure plots the correlation between the average mobility and confirmed cases over the spring of 2020. Mobility data comes from Google's cellphone-location data (see: notes for Fig 1).

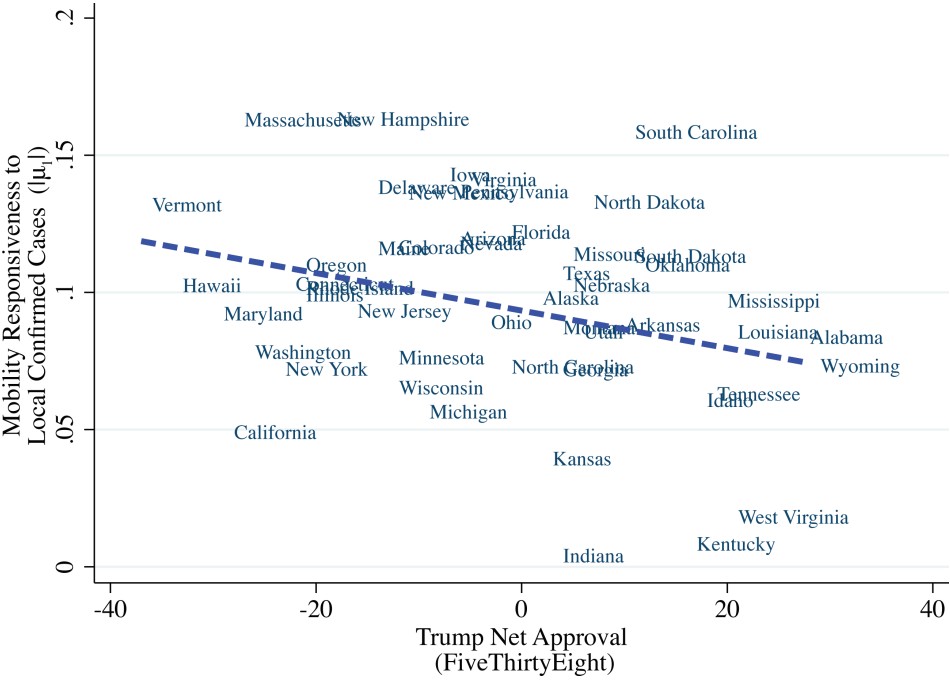

**Fig 6. Mobility responsiveness and presidential approval rating across states.** *Note:* The vertical axis is the state-specific estimate of the coefficient $\mu_1$ of log confirmed cases with mobility as the dependent variable from equation (1). The horizontal axis is the approval rating for the President from FiveThirtyEight (https://github.com/fivethirtyeight/data/tree/master/trump-approval-ratings) averaged over the spring of 2020.

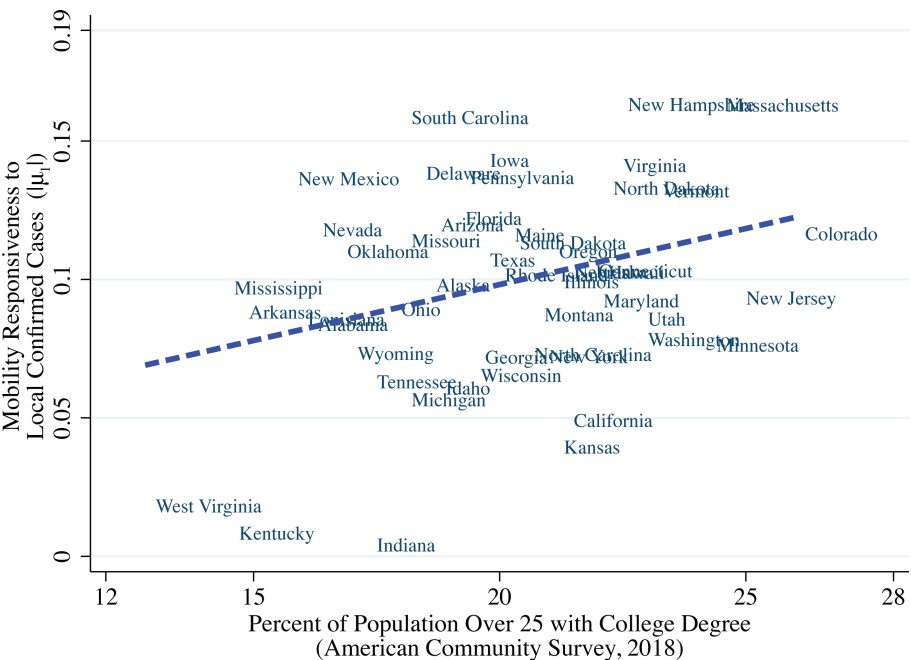

**Fig 7. Mobility responsiveness to confirmed cases and President's approval rating.** *Note:* The vertical axis is the state-specific estimate of the coefficient $\mu_1$ of log confirmed cases with mobility as the dependent variable from equation (1). The horizontal axis is the approval rating for the President from FiveThirtyEight (https://github.com/fivethirtyeight/data/tree/master/trump-approval-ratings) averaged over the spring of 2020.

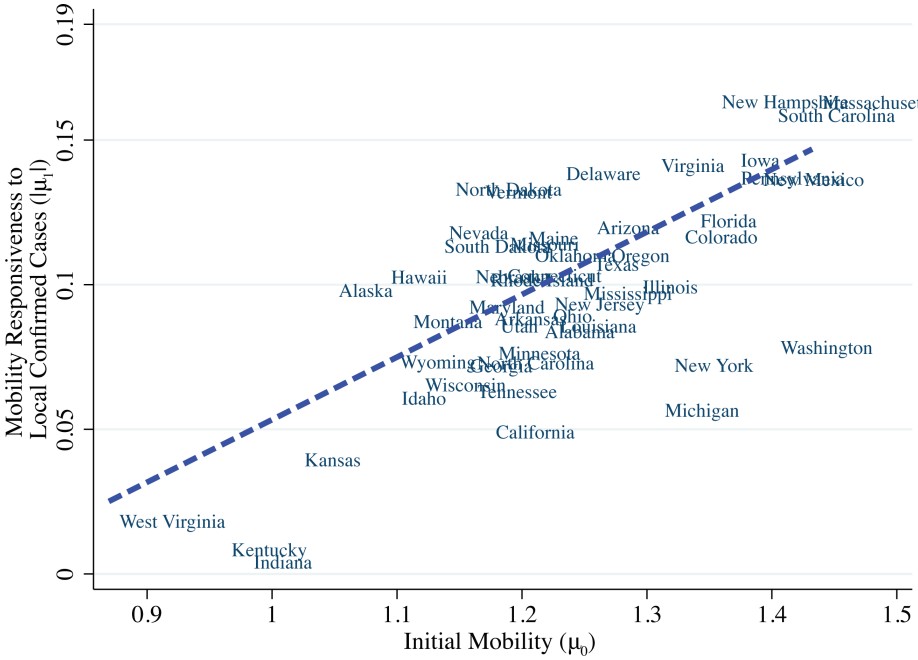

**Fig 8. Mobility responsiveness to confirmed cases and initial mobility.** *Note:* The vertical axis is the state-specific estimate of the coefficient $\mu_1$ of log confirmed cases with mobility as the dependent variable from equation (1). The horizontal axis is the state-specific initial mobility, the constant $\mu_0$ from equation (1) with mobility as the dependent variable.

low and low mobility when cases are high. Similarly, this suggests that a lack of trust in the reported case data could lead to low initial and average mobility because households reduce mobility out of precaution. These figures suggest that to understand voluntary social distancing, we must understand why initial mobility ($\mu_0$) and the response of mobility to information ($\mu_1$) are systematically related.

This reduced form evidence motivates two questions that can only be addressed by a structural model. First, what is driving the patterns of mobility responses $\mu_{s,0}$ vs. $\mu_{s,1}, \mu_{s,2}$, and what do they tell us about information-based social distancing? Second, what are the quantitative implications of the reduced form evidence for the effectiveness of lockdown-induced as opposed to information-based social distancing in combating COVID-19?

## 2 Theory and empirical approach

### 2.1 Model

#### 2.1.1 Basic structure

Our starting point is the "compartmental" disease model as in [4], with recent extensions allowing for social distancing, see [11], [12], and [13]. The total population can be compartmentalized according to

$$S_t + E_t + I_t + R_t + F_t + C_t = N, \tag{2}$$

with the following groups in temporal order of the disease progression

- $S_t$: Susceptible
- $E_t$: Exposed to the virus but not yet infected and not yet infectious
- $I_t$: Infected and infectious, i.e. possibly displaying symptoms and spreading the virus
- $R_t$: Resolving: fully symptomatic and moving towards recovery or death
- $F_t$: Fatalities
- $C_t$: ReCovered

We include three additional compartments to the basic model, which only includes susceptible, infected, and removed, to match the COVID-19 setting. First, we include the exposed compartment that designates people exposed to the virus, and that will eventually get sick but are not yet showing symptoms. This compartment is consistent with evidence on the incubation of the virus during the first week of exposure and allows us to capture one of the benefits of proactive testing. Namely, random testing or contact tracing can potentially find exposed people and quarantine them before they can further spread the virus.

Second, following [12], we add the resolving, fatality, and recovered compartments to capture fatality dynamics. The recovered compartment can later also be used to flow back into the pool of susceptible persons if an immunity to COVID-19 turns out to be only temporary.

#### 2.1.2 Asymptomatic transmission

An important mechanism for the spread of COVID-19 is the possibility that asymptomatic people are still infected and contagious. For instance, evidence from the COVID-19 outbreak on the Diamond Princess cruise ship suggests that around 18% of infected cases were asymptomatic, see [14]. The possibility of asymptomatic exposure affects disease dynamics in at least two ways. First, asymptomatic infectious people worsen contagion and accelerate the spread of the virus. Second, people infected but never display any symptoms will also never face the risk of dying but will eventually contribute to herd immunity. Additionally, we are very

aware that the evidence from the Diamond Princess suffers from sample selection in terms of age and other demographics. For example, [15] document that almost 60% of passengers on the Diamond Princess were older than 60. Therefore, instead of calibrating the probability of being asymptomatic, we directly model asymptomatic infected people and separately estimate the probability of an infected person to not exhibit any symptoms with the parameter $\alpha$. As emphasized by [16], this parameter is also key in estimating a disease model from available time-series data.

Specifically, in the model, the possibility of an asymptomatic infection enters in the stages after the initial exposure. Asymptomatic infections are assumed to have the same transmission rate as symptomatic infections and will have the same duration of infectious and resolving states, but will never result in death. While we are aware that this is potentially a strong assumption, we also point out that it is straightforward to relax it to allow for lower transmission rates by asymptomatic individuals.

### 2.1.3 Testing, information states and sample selection

The difference between symptomatic and asymptomatic COVID-19 infections also matters for the detection of cases through testing. Specifically, symptom-based testing cannot detect asymptomatic infections and, therefore, is unable to reduce contagion through asymptomatic people. Furthermore, since only people in infected and resolving stages $I_t, R_t$ display symptoms, symptom-based testing cannot detect exposed, pre-symptomatic cases in $E_t$. We contrast symptom-based testing with proactive testing, which includes random testing as well as contact-tracing. Proactive testing can detect cases that have been exposed, as well as asymptomatic infections. We summarize the possible information states in Fig 9. These four information states apply to the infected and resolving stages, which we keep track of separately. In other words, for both infected and resolving cases, there will be four sub-states, corresponding to undetected symptomatic, detected symptomatic, detected asymptomatic, and undetected

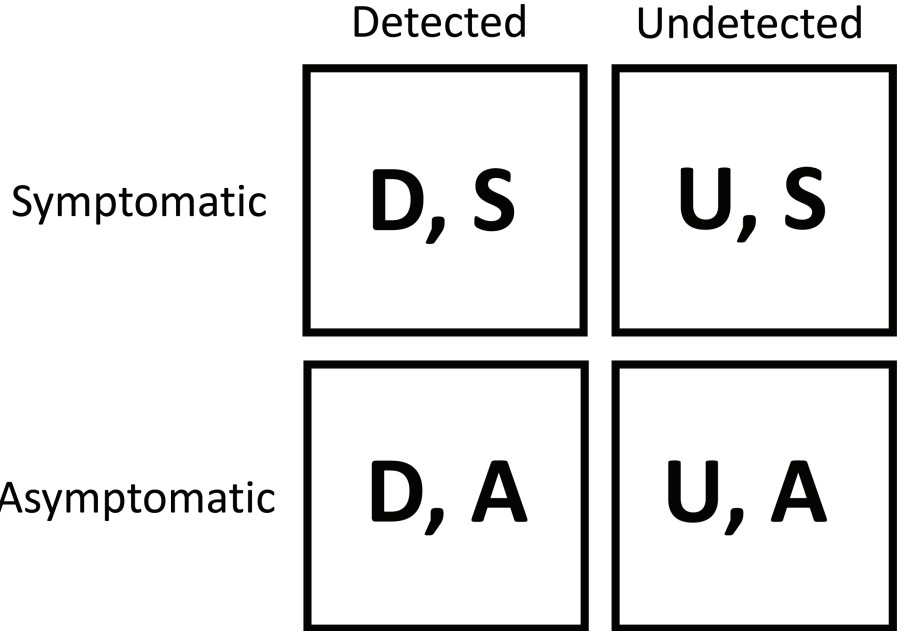

**Fig 9. Information states in the model.**

asymptomatic cases. We also assume that conditional quarantine works perfectly for detected cases so that people who know they tested positive for COVID-19 promptly self-quarantine.

We allow for time-varying testing rates to capture the fact that testing capabilities across states increased over time. To fix ideas, let $k \in \{S, P\}$ denote either symptom-based or proactive testing and assume that testing is initially detecting infected people at a rate $\tau_{k,0}$ and is increasing to a final level of $\tau_{k,1}$. We assume that the increase in testing capability follows a smooth exponential transition with transition rate $\eta_k$ for $k \in \{S, P\}$

$$\tau_{k,t} = \tau_{k,0} \cdot \exp\{-\eta_k \cdot t\} + \tau_{k,1} \cdot (1 - \exp\{-\eta_k \cdot t\}). \qquad (3)$$

Our estimation strategy will then estimate the parameters $\tau_{k,0}, \tau_{k,1}, \eta_k$ for $k \in \{S, P\}$ separately for each state.

## 2.2 Dynamic system

We formalize the ideas of asymptomatic disease transmission and sample selection through symptom-based testing in the following dynamic system. For an overview of the notation of the different compartments, see the flowchart in Fig 10. Following our discussion in Section 2.1.3, we use the notation $i,j$, where $i \in \{D, U\}$ for "detected" and "undetected" cases and $j \in \{S, A\}$ for "symptomatic" and "asymptomatic" cases respectively.

### 2.2.1 Exposure stage

Following the SEIR literature, we assume random matching of infectious and susceptible people which gives the following definitions of the change in susceptible, exposed, and exposed detected people,

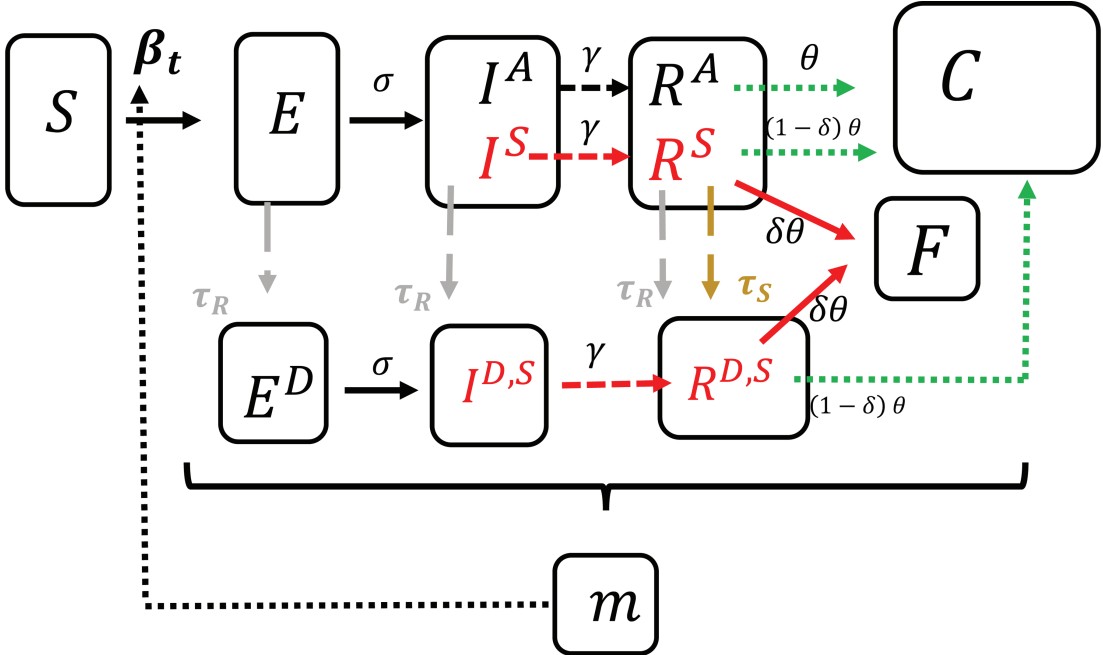

**Fig 10. Flow chart of model.**

$$\Delta S_{t+1} = -\beta_t \cdot \frac{I_t^U \cdot S_t}{N} \tag{4}$$

$$\Delta E_{t+1} = \beta_t \cdot \frac{I_t^U \cdot S_t}{N} - \sigma \cdot E_t - \tau_{P,t} \cdot E_t \tag{5}$$

$$\Delta E_{D,t+1} = \tau_{P,t} \cdot E_t - \sigma \cdot E_{D,t}. \tag{6}$$

In these first stages, people move from susceptible to exposed through contact with infected people, and after an incubation period, they move into the infectious stage at rate $\sigma$. Equations (4) to (6) formalize two points in particular. First, susceptible people can only be exposed to the virus by undetected infectious people $I^U$. In this sense, proactive testing and quarantining will reduce the pool of undetected infectious people and slow the disease spread. Second, proactive testing reallocates people from the group of undetected exposed people to detected exposed people. However, since exposed people are by definition not symptomatic yet, symptom-based testing does not change anything at this stage.

Additionally, a key innovation of our model is the way we allow time variation in disease transmission rates $\beta_t$. Specifically, we assume that

$$\beta_t = \beta_0 \cdot m_t^\psi. \tag{7}$$

In other words, disease transmission is driven by the way randomly matched people interact with each other, captured by the variable $m_t$, which denotes mobility. Lower mobility $m_t$ corresponds to a higher degree of social distancing, which will slow the disease spread. Importantly, we allow the effectiveness of social distancing efforts to vary by location through the parameter $\psi$, which would also capture variances in the efficacy of distancing due to, for example, masking efforts. A natural benchmark for this parameter is $\psi = 2$, which corresponds to social distancing being proportional to the random matching technology given in equation (4). In other words, if $\psi = 2$ people are randomly matched and the mobility choices of people have a proportional impact on disease transmission.

Note that by allowing for variation in curvature $\psi$ in the matching function provides a wide range of behaviors. For example, this allows for the interpretation that the most infectious behaviors are done first and that subsequent increases in mobility affect disease spread less. That's as if the curvature of the matching function is less than quadratic ($\psi < 2$), since the most infectious behaviors spread the disease the most and as mobility increases further, infectiousness of the virus increases less. If on the other hand, the most infectious behaviors are marginal, then the curvature of infections becomes more convex than quadratic, i.e. $\psi > 2$. Larger values of $\psi$ will capture increased transmission, for example, through people meeting at super-spreading events such as choir practice, weddings, concerts, etc.[7]

It should also be noted that $\psi$ will play a dual role in our model. On the one hand, higher values of $\psi$ imply stronger negative health externalities from spreading the disease, which we discuss in the context of individually optimal mobility choices below. On the other hand, higher values of $\psi$ lead to a more aggressive spread of the virus. This also implies that higher values of $\psi$ make social distancing more effective in slowing down the disease's spread. We will return to this issue in our discussion of results.

### 2.2.2 Infectious stage

After an initial incubation period, people become infected and infectious. Since at this stage exposed people can become symptomatic, we start tracking different health and information states, as discussed in Section 3.1.3. People arrive at rate $\sigma$ in the infectious stage after going through the post-exposure incubation period. Of these arrivals, a fraction $\alpha$ will be asymptomatic, while a fraction $1 - \alpha$ will display symptoms. Together, this produces the equations for the change in infectious people that are detected or undetected (denoted by a D or U superscript) and symptomatic or asymptomatic (denoted by an S or A superscript)

$$\Delta I_{t+1}^{U,A} = \alpha \cdot \sigma \cdot E_t - \gamma \cdot I_t^{U,A} - \tau_{P,t} \cdot I_t^{U,A} \tag{8}$$

$$\Delta I_{t+1}^{U,S} = (1 - \alpha) \cdot \sigma \cdot E_t - \gamma \cdot I_t^{U,S} - \tau_{P,t} \cdot I_t^{U,S} - \tau_{S,t} \cdot I_t^{U,S} \tag{9}$$

$$\Delta I_{t+1}^{D,A} = \alpha \cdot \sigma \cdot E_{D,t} - \gamma \cdot I_t^{D,A} + \tau_{P,t} \cdot I_t^{U,A} \tag{10}$$

$$\Delta I_{t+1}^{D,S} = (1 - \alpha) \cdot \sigma \cdot E_{D,t} - \gamma \cdot I_t^{D,S} + \tau_{P,t} \cdot I_t^{U,S} + \tau_{S,t} \cdot I_t^{U,S}. \tag{11}$$

The infectious stage also shows how testing and quarantining impact disease spread. We assume that compliance with quarantining after an individual has tested positive for COVID-19 is perfect and we loosen this strict interpretation in the calibrated model, which we return in the next paragraph.

Since people can display symptoms, both proactive and symptom-based testing will reallocate people from being undetected to detected cases. Detection of cases here matters, since detected cases will be quarantined and therefore not contribute to the spread of the disease in equation (4), since $I_t^U = I_t^{U,A} + I_t^{U,S}$. However, it should be noted that even here, proactive and symptom-based testing differ. Symptom-based testing only detects cases in the fraction $1 - \alpha$ of the infectious population that actually displays symptoms, so reallocates from equation (9) to equation (11). In contrast, proactive testing additionally reallocates cases from undetected asymptomatic to detected asymptomatic cases, i.e. from equation (8) to equation (10).

Although we assume perfect compliance with quarantining after testing positive, it would be straightforward to extend the model to allow for imperfect quarantine compliance. This would mainly entail allowing a fraction of positively tested people to flow back into the pool of infectious people. We do not pursue this route here, since it is plausible that imperfect compliance cannot be separately identified from imperfect detection through testing. As a result, one should interpret our estimated testing effectiveness parameters $\tau$ as a combination of testing effectiveness and imperfect quarantine compliance.

### 2.2.3 Resolving stage

In this penultimate stage, people stop being infectious at rate $\gamma$ and start transitioning into the final stages at rate $\theta$. As before, we need to keep track of four state variables associated with the differences in case detection and case symptoms.

$$\Delta R_{t+1}^{U,A} = \gamma \cdot I_t^{U,A} - \theta \cdot R_t^{U,A} - \tau_{P,t} \cdot R_t^{U,A} \tag{12}$$

$$\Delta R_{t+1}^{U,S} = \gamma \cdot I_t^{U,S} - \theta \cdot R_t^{U,S} - \tau_{P,t} \cdot R_t^{U,S} - \tau_{S,t} \cdot R_t^{U,S} \tag{13}$$

$$\Delta R_{t+1}^{D,A} = \gamma \cdot I_t^{D,A} - \theta \cdot R_t^{D,A} + \tau_{P,t} \cdot R_t^{D,A} \tag{14}$$

$$\Delta R_{t+1}^{D,S} = \gamma \cdot I_t^{D,S} - \theta \cdot R_t^{D,S} + \tau_{P,t} \cdot R_t^{U,S} + \tau_{S,t} \cdot R_t^{U,S} \tag{15}$$

with $R_t^A = R_t^{U,A} + R_t^{D,A}$.

This resolving stage is important for several reasons. First, it helps us match the time-delay from confirmed case data to fatality data by calibrating the associated transition rate $\theta$. Second, testing at this stage dilutes the effectiveness of proactive and symptom-based testing in uncovering disease spread. Recall that cases uncovered in this late resolving stage actually have stopped being infectious, so no longer spread the disease. Third, these cases will still be detected and therefore contribute to the publicly disclosed case count. As a result, susceptible but uninfected people will tend to more aggressively socially distance in response to these higher case counts.

### 2.2.4 Final stage

Arrival in the final stage results in one of two possible outcomes: recovery or death. To simplify our analysis, we assume that both detected and undetected cases have an identical chance of dying $\delta_t$.[8] The basic idea behind this assumption is that irrespective of detection, people might eventually check themselves into a hospital at some point in the resolving stage and therefore get treatment. Death rates therefore measure fatality rates net of treatment effects at the hospital. The resulting number of recovered cases is therefore

$$\Delta C_{t+1} = \theta R_t^{U,A} + (1 - \delta_t) \cdot \theta \cdot R_t^{U,S} \tag{16}$$

$$\Delta C_{D,t+1} = \theta R_t^{D,A} + (1 - \delta_t) \cdot \theta \cdot R_t^{D,S}, \tag{17}$$

with the number of fatalities given by

$$\Delta F_{t+1} = \delta_t \cdot \theta \cdot \left( R_t^{U,S} + R_t^{D,S} \right). \tag{18}$$

We also allow for time-varying death rates, which capture improvements in COVID-19 therapies and are consistent with the divergence in the data between fatality count and confirmed case counts. The dynamic fatality rate is given by

$$\delta_t = \delta_0 \cdot \exp\left\{ -\eta_\delta \cdot t \right\} + \delta_1 \cdot \left( 1 - \exp\left\{ -\eta_\delta \cdot t \right\} \right), \tag{19}$$

were we impose $\delta_1 \leq \delta_0$ in our estimation.

### 2.2.5 Publicly Observable Information

We assume that state health officials immediately disclose detected cases to the public. For fatality counts, we assume that all COVID-19 related deaths are correctly counted, irrespective of whether these were actually detected cases or not. This is consistent with the practice of adding "probable COVID-19 deaths" to the confirmed COVID-19 fatalities. The number of confirmed COVID-19 cases, in contrast, will depend on the state-specific testing regime. For example, suppose a state only uses symptom-based testing, as was widely the case especially in the early stages of COVID-19. Then the observable confirmed case count is given by

$$O_t = I_t^{D,S} + R_t^{D,S}. \tag{20}$$

With symptom-based testing, only infectious and resolving people with symptoms can actually be detected and therefore part of the observable confirmed case counts. In contrast, proactive testing and contact tracing imply the following confirmed case count:

$$O_t = I_t^{D,S} + R_t^{D,S} + E_{D,t} + I_t^{D,A} + R_t^{D,A}. \tag{21}$$

In addition to detecting cases in the pool of symptomatic people, proactive testing enables detection in the groups of exposed and asymptomatic infectious as well as resolving cases.

In the data, observed confirmed case counts will be a function of the mix of symptom-based and proactive testing and will vary over time, which is why we will estimate the associated testing capability parameters, as discussed in section 3.1.3. These public disclosures will then feedback into disease transmission through the voluntary social distancing decisions of individuals.

### 2.2.6 Voluntary social distancing, based on public information disclosures

Our model of voluntary social distancing builds on the framework by [17] for mobility choices in the face of risky COVID-19 infection. We analyze mobility choices from the perspective of a representative person, who thinks that they are uninfected. Let $m_t \in [0, 1]$ denote mobility-based economic activities, such as going out to work, grocery shopping, visiting restaurants and bars, etc. Mobility provides a direct flow utility of $u(m_t)$. In choosing to what extend to involve in mobility-based economic activities, people consider two possible health states $S$ and $E$. If they stay susceptible, then the continuation value is given by $V(S)$, while becoming exposed to the virus and therefore infected and ultimately the possibility of death is captured in the value function $V(E)$, with $V(E) < V(S)$. We assume that people perceive the probability of being exposed to the virus as a simple linear function in their mobility choices $\tilde{b}_t \cdot m_t$, where $\tilde{b}_t$ is the perceived probability of being exposed to the virus per unit of mobility. The optimal mobility choice then obeys the following Bellman equation

$$V(S) = \max_{m_t} \ u(m_t) + \phi \cdot \left[ \tilde{b}_t \cdot m_t \cdot V(E) + (1 - \tilde{b}_t \cdot m_t) \cdot V(S) \right]. \tag{22}$$

where $\phi$ denotes a discount factor. The implied first order condition for the mobility choice is therefore

$$u'(m_t) = \tilde{b}_t \cdot \phi \cdot \left[ V(S) - V(E) \right]. \tag{23}$$

In other words, people optimally weight the marginal benefit of mobility-based economic activities against the possible continuation value loss from becoming infected. We note that since our model will be estimated on a daily frequency, the continuation values $V(S)$ and $V(E)$ are unlikely to be time-varying. Since mobility is a static policy variable, the key to determining the extent of mobility is the expected infection probability $\tilde{b}_t$.

Under the random matching in equation (4) and non-linear social interaction in equation (7), rational expectations imply that

$$\bar{b}_t = \beta_0 \cdot \left( \bar{m}_t^I \right)^{\psi-1} \cdot \left( \frac{I_t^U}{N} \right). \tag{24}$$

Note that since people take this infection probability $\bar{b}_t$ and its component $\left( \bar{m}_t^I \right)^{\psi-1}$ as given when optimizing the Bellman equation (22), they ignore any externality they impose on other people by increasing their mobility. Importantly, the strength of this health externality is governed by the contagiousness of interactions $\psi$. As social interactions become more contagious (e.g., people attend more super-spreading events, such as choir practice, concerts, weddings, etc.) $\psi$ increases, implying more negative health externalities from the spread of COVID-19.

 

A natural solution concept for infection probability $\bar{b}_t$ might be Rational Expectations or Nash equilibrium, under which people correctly forecast $b_t$. However, this would require people to correctly forecast the unobservable variables $\left(\bar{m}_t^I\right)^{\psi-1}$ and $\left(\frac{I_t^U}{N}\right)$. There are several reasons why this solution concept might be considered problematic for modeling expectation formation during a pandemic. First, the non-stationarity of disease dynamics in the short run—especially in the daily frequency data we consider—is likely to prevent the convergence of simpler expectations, such as adaptive expectations to rational expectations as in [18]. Second, [19] argues that even in deterministic non-stationary environments, simple extrapolation forecasts can outperform unbiased rational expectations based on the true model and might therefore be preferred.

We model expectations formation via simple Bayesian updating. We assume every morning, people wake up and form an expectation on infection probability if being mobile that day, based on their prior and newspaper reports of confirmed cases and fatalities. Focusing on $\ln \tilde{b}_t$ as the log expected infection probability belief at time $t$, we assume that the prior is normally distributed, $N\left(\ln(b_0), \sigma_0^2\right)$. People also consider the combined signal

$$X_t = \nu_1 \cdot \ln O_t - \nu_2 \cdot \ln F_t, \tag{25}$$

with $\nu_1 > 0$ and $\nu_2 > 0$. In other words, people use the observed count of confirmed COVID-19 cases $O_t$, and the total cumulative fatality count $F_t$ to predict the probability of getting exposed to the virus per unit of mobility. This shows that the perceived infection probability will increase in the number of observed cases $O_t$, as people predict that the likelihood of running into an infected person is higher with higher case counts. At the same time, people believe that an increase in the total fatality count will tend to decrease the infection probability as it is not possible to run into dead people. We assume that the combined signal $X_t$ perceived to be normally distributed with $N\left(\ln(b^*), \sigma_\epsilon^2\right)$. Note that $\ln(b^*)$ could be the correct log average infection rate in the case of rational expectations, but we do not take a stand on it here. More importantly, $\sigma_\epsilon^2$ captures the variance of noise in the signal. The higher this noise, the less credible people think the publicly provided information is.

The posterior can therefore be written as

$$\ln \tilde{b}_t = \rho_\epsilon \cdot \ln b_0 + (1 - \rho_\epsilon) \cdot \nu_1 \cdot \ln O_t - (1 - \rho_\epsilon) \cdot \nu_2 \cdot \ln F_t, \tag{26}$$

where $\rho_\epsilon = \frac{\sigma_\epsilon^2}{\sigma_\epsilon^2 + \sigma_0^2} \in (0, 1)$ is the belief on the importance of noise in the data. Higher values of $\rho_\epsilon$ therefore increase the weight in the prior belief of the infection rate, while weight on the publicly published data is reduced. On the other hand, if public information is very credible, $\rho_\epsilon$ will be very low, therefore placing less weight on the prior infection probability and making expectations more responsive to published case and fatality counts.

For our empirical implementation, we combine this log-linear expectation formation in equation (26) with exponential utility for the flow utility from mobility:

$$u(m_t) = u_0 - \exp\{-\kappa \cdot m_t\} \tag{27}$$

with $\kappa > 0$. As a result, the first-order condition in equation (23) combined with equations (27) and (26) can be rewritten to our empirical equation:

$$m_t = \mu_0 + \mu_1 \cdot \ln O_t + \mu_2 \cdot lnF_t - \lambda_t \tag{28}$$

the constants are given by

 

- $\mu_0 = \rho_\epsilon \cdot \left(-\frac{\ln b_0}{\kappa}\right) - \frac{1}{\kappa} \cdot \ln\left(\frac{\phi}{\kappa}[V(S) - V(E)]\right)$, which captures unconditional mobility, irrespective of public information. Note that $\frac{\partial \mu_0}{\partial b_0} = \rho_\epsilon \cdot \left(-\frac{1}{\kappa \cdot b_0}\right) < 0$, so higher prior beliefs on infection $b_0$ reduce mobility.
- $\mu_1 = -(1 - \rho_\epsilon) \cdot \frac{\nu_1}{\kappa} < 0$ is the mobility response to confirmed case counts
- $\mu_2 = (1 - \rho_\epsilon) \cdot \frac{\nu_2}{\kappa} > 0$ is the mobility response to cumulative fatality counts.

Equation (28) captures the reduced form social distancing behavior. Importantly the parameter $\mu_1$ is that [7] refers to as "prevalence elasticity", that is the response of risky behavior to prevalence. Equation (28) also helps us to understand the reduced form regression of equation (1) better and the associated results in Fig 8. Expectation formation parameters $\rho_\epsilon$, $b_0$, $\nu_1$, and $\nu_2$ enter the mobility coefficients $\mu_0$, $\mu_1$, and $\mu_2$. If preference parameters $\kappa$ and $\phi$ as well as expectation coefficients $\nu_1$ and $\nu_2$ are similar across states, average mobility $\mu_0$ and mobility responses to new information $\mu_1$ and $\mu_2$ are directly informative about how much people trust the quality of public disclosures in their states, or $\rho_\epsilon$. Low information quality (or high $\rho_\epsilon$) will translate into low responsiveness ($\mu_1$ and $\mu_2$), while the opposite is true for high information quality (low $\rho_\epsilon$). At the same time, low information quality (or high $\rho_\epsilon$) will tend to reduce initial mobility $\mu_0$. As a result, information quality can explain the relationship between initial mobility and prevalence elasticity in 8. This captures the idea people in states with low information quality assume such low-quality information is a signal that news is bad, i.e., infection probabilities are high, which is why their initial mobility $\mu_0$ as well as the absolute value of their prevalence elasticity $\mu_1$ is low.[9]

Information policies across locations will impact the parameter $\rho_\epsilon$, which in turn influences the parameters $\mu_0, \mu_1, \mu_2$. This relation suggest that we can get an idea of how changes in information policies change the propagation of the virus, by imposing different values of $\mu_0, \mu_1, \mu_2$, which will be the core of our quantitative analysis of information policies.

We also add the term $\lambda_t$ to model the effects of (temporary) lockdown-induced social distancing. These are captured by the following time-varying variable as in [20]:

$$\lambda_t = \lambda_0 \cdot \exp\{-\eta_L \cdot t\} + \lambda_1 \cdot (1 - \exp\{-\eta_L \cdot t\}), \tag{29}$$

where we estimate the parameters $\lambda_1 > \lambda_0$ and $\eta_L$ from the data.

### 2.3 Structural estimation

We proceed in three steps to estimate our model. In the first step, we estimate equation (28) directly using data on mobility, observed case counts, cumulative fatalities, and state-level lockdowns to estimate the parameters ($\mu_0$, $\mu_1$, $\mu_2$, and $\lambda_1$) separately for each state. In our second step, we calibrate three "clinical" parameters that capture important stages of disease progression and for which we believe there is convincing evidence from the micro-data. We start by setting the initial virality $R_0$ of COVID-19 to 6 based on evidence by [21] on the spread of COVID-19 during the early phases of the outbreak in Wuhan, China. However, it is important to note that the model will estimate time variation in the virality $R_t$, taking endogenous social distancing into account,

$$R_t = \beta_t \cdot \frac{1}{\gamma}. \tag{30}$$

Virality at time $t$ is the product of $\beta_t$, which is a function of mobility $m_t$ and the duration of cases remaining infectious $1/\gamma$, which will also be estimated. The second calibrated parameter is the average incubation period, which we set to 5 days, so that $\sigma = 1/5$. This parameter value

is consistent with the evidence in [22]. The third calibrated parameter is the average resolution time, which we set to 12 days, so that $\theta = 1/12$, based on evidence from [23]. Of course, our specific quantitative results depend on these calibrated numbers. Future researchers can use different calibrations based on future studies.

In our third step, we estimate the remaining parameters using Simulated Methods of Moments, see [24]. Specifically, we choose six testing parameters ($\tau_{P,0}, \tau_{P,1}, \eta_P, \tau_{S,0}, \tau_{S,1}$, and $\eta_S$), five initial values of undetected cases ($E_1, I_1^{U,S}, I_1^{U,A}, R_1^{U,S}$, and $R_1^{U,S}$) and six parameters related to disease transmission and fatalities ($\alpha, \delta_0, \delta_1, \eta_\delta, \gamma$, and $\psi$). These 17 parameters are chosen to minimize

$$SSE = \sum_{t=1}^{T_{in}}(O_t - O_t^M)^2 + \sum_{t=1}^{T_{in}}(F_t - F_t^M)^2 + \sum_{t=1}^{T_{in}}(m_t - m_t^M)^2, \tag{31}$$

where $O_t^M, F_t^M$, and $m_t^M$ are model-generated time paths while $O_t, F_t$, and $m_t$ are the corresponding data time paths. As equation (31) shows, we match three time paths: observed confirmed cases, cumulative fatalities, and mobility.

Our estimation is also subject to three inequality constraints:

$$\tau_{P,0} \leq \tau_{P,1} \tag{32}$$

$$\tau_{S,0} \leq \tau_{S,1} \tag{33}$$

$$\delta_0 \geq \delta_1 \tag{34}$$

and 17 variable bounds, ensuring that transition rates remain $\in (0,1)$ and initial numbers of undetected infections are non-negative. We also utilize additional micro-evidence to bound parameter values for several key parameters. First, we impose an upper bound of death rates for symptomatic people of 15%, consistent with the case fatality rate of 15%, which prevailed in Italy at the height of the COVID-19 crisis in that country. Italy's case fatality ratio, in turn, is the highest currently reported case fatality ratio in the world. Second, we bound the probability of being asymptomatic, conditional on infection, to values $\alpha \in [0.05, 0.8]$. The lower bound corresponds to the fraction of patients in [25] who never developed symptoms, while the upper bound corresponds to the upper bound of estimates for $\alpha$ in randomized testing data in [26]. Most of the existing estimates for $\alpha$ comfortably fall within these bounds, such as evidence from the Diamond Princess at $\alpha = 0.18$ in [14] and $\alpha = 0.45$ in [27].

We tested the identification of this Simulated Method of Moments estimator in two ways. First, we simulated artificial time paths for a given set of parameters and made sure that the estimation procedure converges to the correct values from random starting points.[10] Second, we cross-checked the parameter estimates with the intuitive co-movement in the data. For example, consider several key parameters: $\psi, \gamma$, and $\alpha$. First, contagiousness of community interactions $\psi$ is in part pinned down by the co-movement between mobility $m_t$ and confirmed cases $O_t$ because $\psi$ directly influences how strongly any social distancing translates into the new infections. Second, the rate $\gamma$ determines how long people stay infectious after the virus has incubated and this parameter estimate is driven by the co-movement of confirmed cases and fatalities. In particular, higher values of $\gamma$ decelerates disease transmission, as it reduces the pool of infected people. At the same time, higher values of $\gamma$ will lead to a faster transition of cases from infectious to resolving and therefore accelerate growth in fatalities. Finally, the probability of being asymptomatic conditional on being infected ($\alpha$) is strongly driven by the shape of the number of confirmed cases. Higher values of $\alpha$ increase disease transmission through asymptomatic people while also strengthening herd immunity since a higher number of asymptomatically infected people recover without symptoms,

which in turn reduces the pool of susceptible people. Furthermore, a higher value of $\alpha$ tends to decelerate fatality counts, as asymptomatic cases all eventually recover. In other words, the shape of the time path of confirmed cases, the timing of the peak in confirmed cases, and the co-movement of confirmed cases with fatalities will be important for pinning down $\alpha$.

In our fourth step, we improve our model's forecast properties by using techniques from Machine Learning. This step is important because complex and non-linear models tend to overfit the data and perform poorly in terms of out-of-sample predictions, see [28]. In turn, poor out-of-sample predictions indicate that model parameters do not fit robust, generalizable patterns but instead idiosyncratic noise in the data. To improve model generality, we use two key ideas from Machine Learning, ensemble learning and cross-validation. Combining different models into an averaged ensemble forecasts, stabilizes the predictions, and tends to reduce the variance of the prediction error. Additionally, cross-validation allows us to compute optimal ensemble weights to maximize out of sample accuracy.

For cross-validation, we reserve the last week days of data as our out-of-sample prediction window. To estimate different models, we re-estimate the model by removing one day at a time, going back 28 days, and use these shortened training samples as estimation data. We then use forecasts from these 28 models to predict the time path of confirmed cases and fatalities in the 7 days after the end of the last training sample. The model weights are then chosen to minimize the following out-of-sample prediction error

$$SSPE = \sum_{l=1}^{L} \left( O_{T_{in}+l} - \sum_{i=1}^{I} w_i \cdot O_{T_{in}+l}^{(i)} \right)^2 + \sum_{l=1}^{L} \left( F_{T_{in}+l} - \sum_{i=1}^{I} w_i \cdot F_{T_{in}+l}^{(i)} \right)^2 . \tag{35}$$

## 3 Results

### 3.1 Case studies of state estimation: Massachusetts and West Virginia

We begin with two specific estimates of states to explain more precisely how our empirical approach works in practice. We selected the two states based primarily on how strong the voluntary social distancing responses in response to confirmed cases were: people in West Virginia had the lowest estimated absolute value of $\mu_1$, while Massachusetts had one of the highest. These two states will also prominently feature in our analysis of different counterfactual information policies below.

#### 3.1.1 State fundamentals

West Virginia is one of the poorest states and smallest states in the US, and relatedly is not very densely populated.[11] Specifically, West Virginia's population density is around 77 persons per square mile with its largest city, Charleston, counting 45,000 residents, or about 2.5% of the total population. Some of these fundamentals, such as lower population count and less density, make West Virginia unlikely to strongly suffer from COVID-19 in terms of health outcomes. On the other hand, voluntary social distancing may be low in West Virginia because it might have a lower belief about the severity of COVID-19; West Virginia has a net approval for President Trump of about +20% and only 20% of the population over 25 years of age has a BA degree. COVID-19 is a potentially serious health threat for West Virginians because 20% of its population is over 65.

In contrast, Massachusetts is one of the wealthiest, largest, and most densely populated states and is therefore at a higher risk from an aggressively spreading contagious disease, such as COVID-19. Massachusetts is home to about 6.8 million people, around 10% of whom live in its largest city, Boston. It is the 29$^{th}$ most densely populated state with approximately 839

persons per square mile. These factors likely increase the potential threat of COVID-19 for the citizens of Massachusetts. On the other hand, information-based social distancing may be aggressive in Massachusetts because it has the highest fraction of college-educated persons, with 43% of the population over 25 holding a BA degree. Additionally, Trump's presidential approval rating is around -28%. Based on the descriptive evidence, we should expect high social distancing levels and low mobility in Massachusetts.

### 3.1.2 Progression of COVID and State government responses

COVID-19 spread to Massachusetts and West Virginia at very different times, while state actions were taken around the same time. Massachusetts declared its first confirmed case of COVID-19 on March 2, 2020, it took another two weeks until West Virginia identified its first COVID-19 case on March 17, 2020. West Virginia was the last state to announce the confirmation of a COVID case publicly. While two weeks seems like a small-time difference, it should be noted that both disease spread and our empirical analysis are conducted at a daily frequency, which implied a substantial difference in timing. Though the arrival of COVID-19 in both states was different, both states imposed state-wide lockdowns on March 24, 2020. This relative delay of the state response in West Virginia indicates a more hesitant approach to lockdowns. It is also mirrored in the fact that West Virginia's reopening started on May 4, approximately two weeks before partial reopening started in Massachusetts.

With these differences in mind, our mobility measures from Section 1, can help us understand how quantitatively different social distancing was in these two states. Overall, average mobility declined by 31% relative to 2019 in Massachusetts as compared to an 11% average mobility decline in West Virginia. These raw differences could be driven by differences in state lockdown policies as compared to voluntary social distancing. Therefore, using our estimates from equation (1), we can calculate the average effectiveness of lockdowns on mobility, or the term $\lambda_t$. This term turns out to be remarkably similar between the two states. While in place in Massachusetts, our estimates suggest that the lockdown reduced mobility by 12% on each day relative to 2019. In comparison, the West Virginia lockdown reduced mobility by 13% on each day. These effects of lockdowns on mobility each day are close to the median effect of 14% across states.[12] However, the lockdown was in place in Massachusetts for about two weeks longer. This longer duration might at least, in part, contribute to a higher overall effect of lockdowns on the spread of the virus.

We now move to the comparison of the raw data in terms of health outcomes. For comparison purposes, we report population-adjusted cumulative fatalities "per 100,000," which is calculated as $\frac{\text{Number of fatalities}}{\text{State population}} \times 100,000$. In terms of raw fatality outcomes, Massachusetts seems to have performed much worse with 188 deaths per 100,000, while West Virginia has performed relatively well with five deaths per 100,000 until the end of June. Of course, these outcomes by themselves are likely to be driven by the fact that Massachusetts is more densely populated, as discussed in Section 3.1.1. Therefore, to evaluate the effectiveness of lockdown policies and voluntary social distancing, we now move to model estimation.

### 3.1.3 Massachusetts and West Virginia: model estimates and social distancing effectiveness

The panels in Fig 11 show model estimates for West Virginia and Massachusetts. The two vertical lines make different dates for including of the training sample. Between the first and the second vertical line, one day is added at a time to the estimation sample and the model is re-estimated on the extended training sample. Past the second line are the observations that

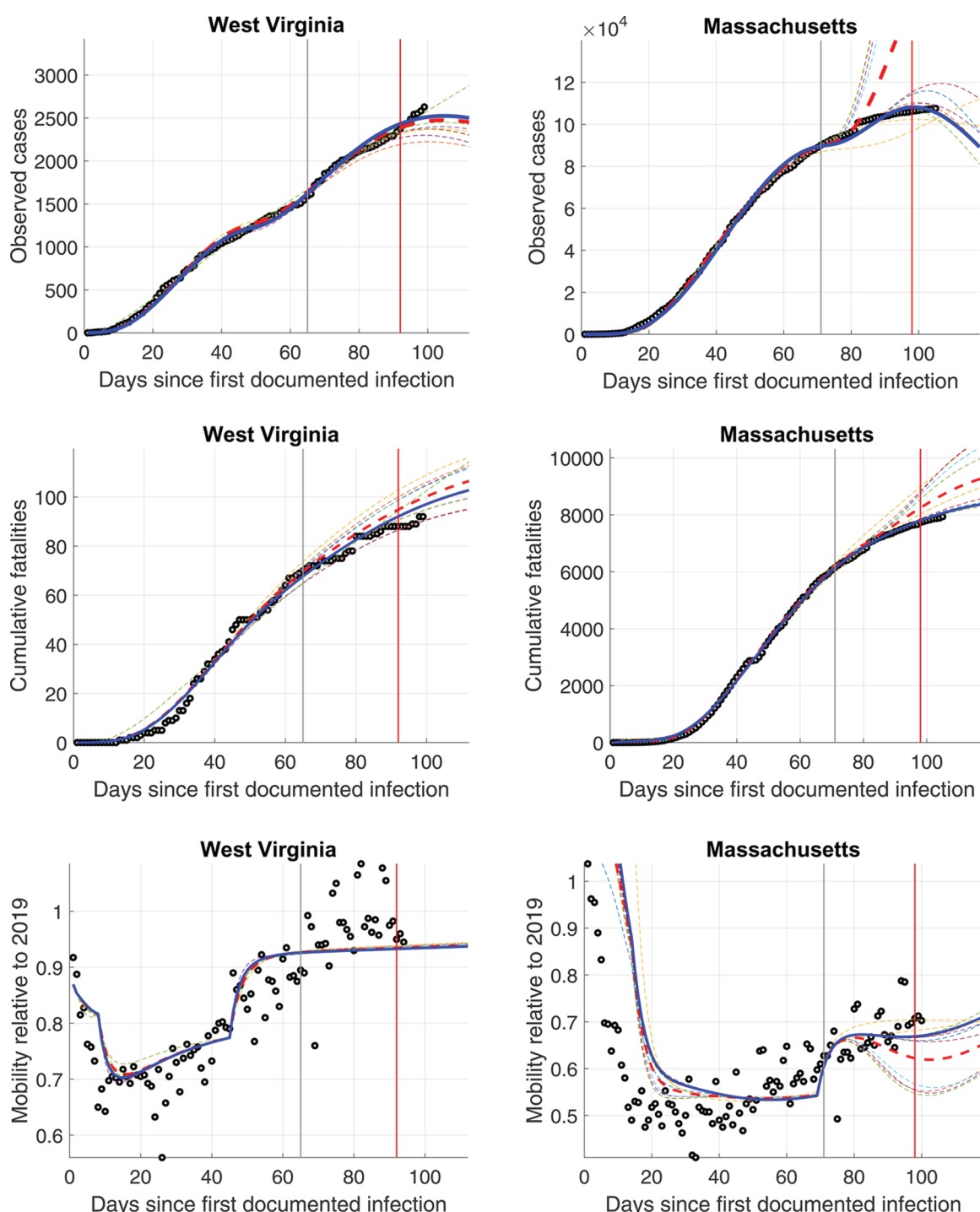

**Fig 11. Model estimates vs data in black dots. First vertical line is end of first training sample, while every day before first and second vertical line is another training sample.** We estimate 28 models, 10 of which are displayed in dashed lines. Data beyond the second line is test data for cross-validation of optimal ensemble model. Optimal ensemble is shown in blue, while naive ensemble, which averages across all 28 models is shown in red dashed line.

constitute the test sample for cross-validation of the ensemble model (see equation (35)). The various dashed lines then show predictions for the first five and the last five models. Our optimal ensemble model estimates are displayed as a solid blue line. For comparison purposes, we also present a naive ensemble in red dashed lines. As the blue lines show, our ensemble estimator successfully predicts the rising number of cases and cumulative fatalities. These figures provide examples for the optimal ensemble, successfully estimating generalizable patterns that go beyond what even a naive ensemble would find. However, note that the model performs somewhat worse in terms of fitting mobility changes, though the R-squared for both states is still around 70-75%.

Once we estimated the optimal ensemble model for both West Virginia and Massachusetts, we turn to the calculation of the causal effects of state lockdowns and voluntary social distancing. The panels in Fig 12 show the qualitative results. In these panels, the blue lines correspond to the optimal ensemble model estimates from Fig 11. Of course, there is uncertainty with these hypothetical conditions, and uncertainty is driven both by model assumptions and calibrations. We contrast this estimated and predicted path with two counterfactuals from the model. First, the grey dashed line is the infection, fatality, and mobility time path without state lockdowns but with voluntary social distancing. Second, the black dotted line displays the same time paths for a counterfactual without voluntary social distancing but with state lockdowns. As the counterfactual panels show, in both West Virginia and Massachusetts, the effect of voluntary social distancing is more important than the impact of state lockdowns: in general, the dotted lines are above the dashed grey lines.

To put these plots in perspective, consider the results in Table 1, which reports results per 100,000. The table shows that state lockdowns were far more effective in Massachusetts than in West Virginia in saving lives. Importantly, even if one adjusts for the fact that the lockdown in Massachusetts had a longer duration than in West Virginia, the state lockdown effectiveness is still twice as high in Massachusetts than in West Virginia. At first, this might seem puzzling, since our estimates of $\lambda_t$ or the effect of lockdowns on mobility were quite similar between the two states. However, recall from equation (7) that how strongly mobility changes depend on the contagiousness of interactions $\psi$. And these parameters differ substantially in our model estimates. For Massachusetts, we estimate an averaged value of $\psi = 2.5$ across models used in the optimal ensemble. This parameter suggests higher effectiveness of social distancing on reducing disease transmission than in the random matching benchmark of $\psi = 2$. In contrast, $\psi = 1.55$ for West Virginia suggests lower effectiveness of social distancing measures there.

These contagiousness differences also magnify already substantial differences in voluntary social distancing. Table 1 shows that voluntary social distancing was more than twice as effective in saving lives in Massachusetts as in West Virginia.

## 3.2 Key parameter estimates for all states

We now move to a more general discussion of our estimation results across states. In this section, we focus on a handful of parameters that prominently featured in the policy discussions around COVID-19. However, before turning to the results, we want to point out an important caveat. We will present "ensemble averages" of parameters, defined as weighted values of parameters for all models in the optimal ensemble, where weights reflect the optimal ensemble weight. These ensemble averages are meant as an easy way to indicate what models estimate. However, it is unlikely that these estimates themselves would exactly give the ensemble models' estimated paths since all these models are highly non-linear. For the ensemble-averaged parameters, we will also mainly focus on the median values, but the reader should

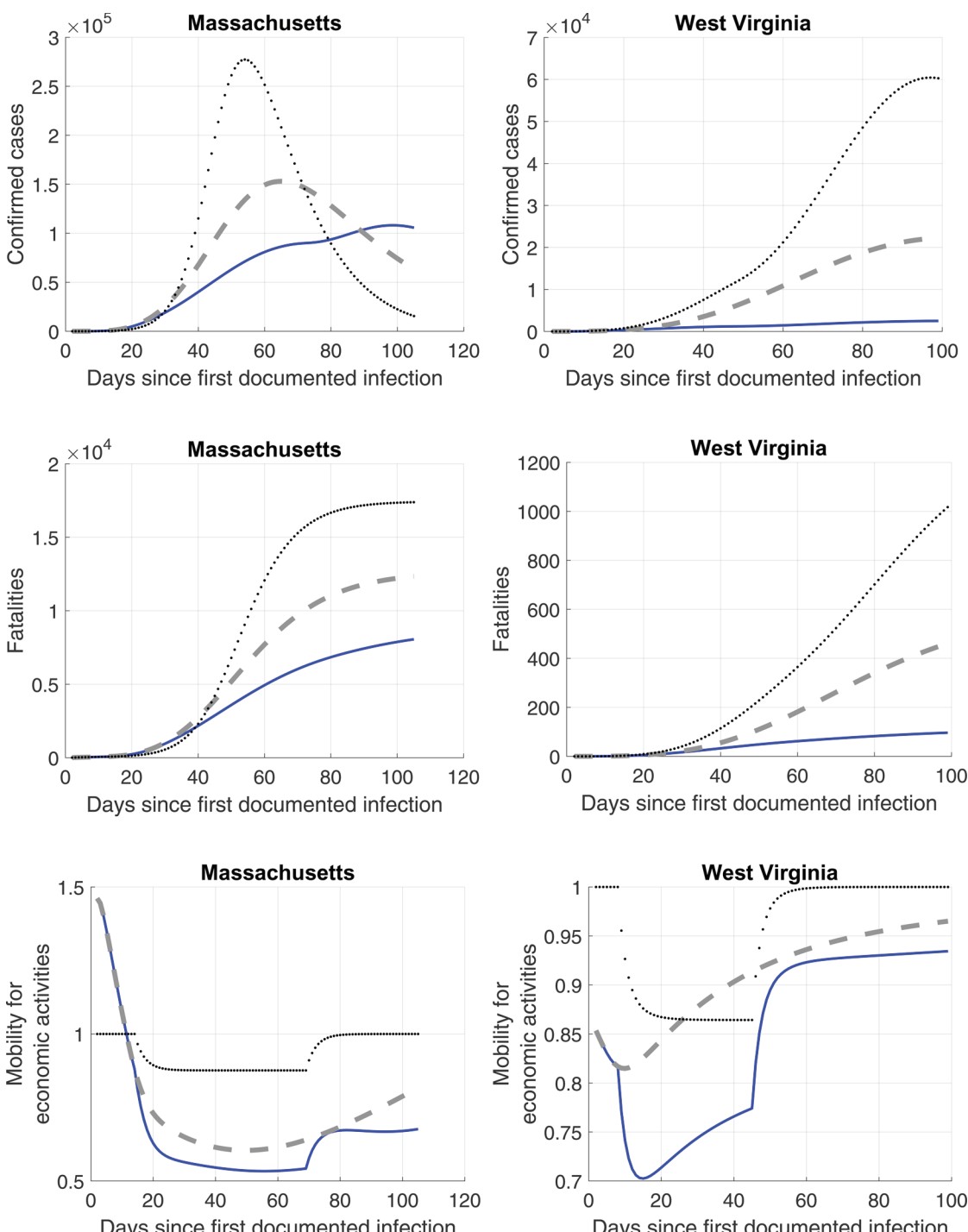

**Fig 12. Blue line captures ensemble estimates from Fig 11.** Dashed line is outcomes without state lockdown but with voluntary social distancing. Dotted line is counterfactual without voluntary social distancing but with state lockdown.

**Table 1. Lives saved by social distancing.**

|                          | West Virginia | Massachusetts |
|--------------------------|---------------|---------------|
| State lockdown           | 20.5          | 61.4          |
| State lockdown[a]        | 20.5          | 42.4          |
| (same duration)          |               |               |
| Voluntary Social Distancing | 51.8       | 133.7         |

*Note:* Entries display population-adjusted number lives saved (per 100,000). It is calculated as
(Number of lives saved/State population) $\times$ 100,000, where "Number of lives saved" is calculated as difference
between counterfactual cumulative fatalities until end of June and estimated cumulative fatalities, both calculated
from the optimal ensemble model (31) and (35).
[a] Lockdown duration is normalized by adjusting lives saved for MA by (38 days/55 days).

**Table 2. Ensemble-weighted parameters.**

|                      | $R_0^a$ | $\bar{R}_t$ | $\tau_{R,0}$ | $\tau_{R,1}$ | $\eta_R$ | $\tau_{S,0}$ | $\tau_{S,1}$ | $\eta_S$ |
|----------------------|---------|-------------|--------------|--------------|----------|--------------|--------------|----------|
| 90th Perc.           | 6.00    | 6.66        | 0.03         | 0.30         | 0.02     | 0.02         | 0.30         | 0.03     |
| 75th Perc.           | 6.00    | 5.12        | 0.00         | 0.20         | 0.00     | 0.00         | 0.29         | 0.02     |
| 50th Perc.           | 6.00    | 3.86        | 0.00         | 0.09         | 0.00     | 0.00         | 0.20         | 0.01     |
| 25th Perc.           | 6.00    | 3.13        | 0.00         | 0.04         | 0.00     | 0.00         | 0.07         | 0.00     |
| 10th Perc.           | 6.00    | 2.66        | 0.00         | 0.01         | 0.00     | 0.00         | 0.02         | 0.00     |

*Note:* Estimates of parameters from model (31), weighted with optimal ensemble weights (35).
[a] Calibrated parameter, using [21].

be aware that extreme values that are estimated quantitatively matter in the non-linear models
considered here.

Turning to Table 2, we point out that average virality $\bar{R}_t$ differs greatly from the calibrated
initial virality estimate of six. These differences are, of course, driven by endogenous social
distancing. Importantly, the median ensemble-average value is 3.86, which is far lower than
six. On the other hand, it is also worth noting that none of our estimates fall below one, which
means that no state has managed to push virality below the net infection growth threshold
persistently.

Our model also provides estimates for the fatality rates due to COVID-19 in Table 3. This
death rate should be interpreted as the death rate for symptomatic people since only symp-
tomatic people can die in our model. The model estimates that in the median state, death rates
for symptomatic people dropped substantially through the estimation period for the median
state. This drop implies an impressive improvement in learning to treat the disease.

The model estimates the probability of being asymptomatic, conditional on being infected
to be around 12% for the median state. That is only a bit lower than the 18% estimated by [14]
for the Diamond Princess. However, it should be noted that estimates for the $\alpha$ parameter
vary from our lower bound of 5% to around 60%, which is still comfortably below our upper
bound of 80%.

We note that $R_0$ and the infected death rate are highly dependent on many factors, includ-
ing the local geography of social networks, demographics, and culture. We also note that
models in the literature have produced different results based on whether they were aggre-
gated (equation-based) or distributed models (e.g., individual or agent-based). These caveats
should be considered when interpreting our results.

### 3.3 The importance of information policy

Table 4 presents results from estimating the effectiveness of different types of social distanc-
ing for all states, using the same methodology as in Section 3.1.3. It shows that in terms of

**Table 3. Ensemble-weighted parameters: Virality and testing.**

|  | $\eta_L$ | $\delta_0$ | $\delta_1$ | $\eta_d$ | $\alpha$ | $\psi$ | $\gamma$ | $\theta$[a] | $\sigma$[b] |
|---|---|---|---|---|---|---|---|---|---|
| 90[th] Perc. | 0.40 | 0.14 | 0.0031 | 0.18 | 0.58 | 5.96 | 0.41 | 0.083 | 0.2 |
| 75[th] Perc. | 0.34 | 0.13 | 0.001 | 0.075 | 0.36 | 4.17 | 0.15 | 0.083 | 0.2 |
| 50[th] Perc. | 0.32 | 0.104 | 0.000031 | 0.036 | 0.12 | 2.53 | 0.083 | 0.083 | 0.2 |
| 25[th] Perc. | 0.31 | 0.034 | 0.0 | 0.025 | 0.051 | 1.91 | 0.049 | 0.083 | 0.2 |
| 10[th] Perc. | 0.26 | 0.014 | 0.0 | 0.017 | 0.05 | 1.06 | 0.036 | 0.083 | 0.2 |

*Note:* Estimates of parameters from model (31), weighted with optimal ensemble weights (35).
[a]Calibrated parameter using [22].
[b]Calibrated parameter using [23].

**Table 4. Estimates of social distancing effectiveness.**

|  | Lockdown | | | Voluntary Social Distancing | | | Econ. cost equiv. |
|---|---|---|---|---|---|---|---|
|  | Lives saved | | Mobility lost | Lives saved | | Mobility lost | Mobility lost given |
| Pct. | Total | Per 100K | Cum. | Total | Per 100K | Cum. | equivalent fatality |
| 90[th] | 78.43 | 6021.87 | 8.26 | 314.86 | 23071.44 | 29.61 | 69.36% |
| 75[th] | 27.93 | 1824.59 | 6.57 | 139.4 | 7283.34 | 20.02 | 19.04% |
| 50[th] | 7.54 | 373.13 | 3.66 | 33.92 | 953.56 | 14.75 | -24.18% |
| 25[th] | 2.66 | 60.31 | 1.95 | 5.73 | 222.62 | 9.12 | -72.66% |
| 10[th] | 0 | 0 | 0 | 1.32 | 9.82 | 7.38 | -89.43 |

*Note:* First column indicates percentile of across state estimates. Estimates are based on difference between actual fatalities or mobility and counterfactual fatalities or mobility without lockdowns (columns 2–4) or without voluntary social distancing (columns 5–7). The last column calculates the percentage mobility loss if lockdowns are used to save the same number of lives as voluntary social distancing in the same state percentile.

total numbers of lives saved, voluntary social distancing was almost three times more effective than state lockdowns for the median state. Effectiveness even increases if we consider population-adjusted lives saved, which implies that voluntary social distancing is 4.5 times more effective. These results suggest that information policy and health advisories, which focus public attention on confirmed cases and fatalities, can be an important tool for policymakers. Indeed, beyond effectiveness in saving lives, information policy tools are attractive because they facilitate private initiative in implementing social distancing. Of course, a major drawback of this argument is that such private initiative can be insufficient in the presence of very strong health externalities, a point to which we return below.

Several factors drive the result that information-based voluntary social distancing has been more effective in saving lives than state lockdown. One of the key factors is that voluntary social distancing tended to sharply depress mobility early on, consistent with the evidence by [10], [29], and Fig 1. In this context, it should be noted that the earlier social distancing is, the more effective it is in reducing the spread of the disease. At the same time, state lockdowns tended to be weeks after the first confirmed cases. Furthermore, most lockdowns in Spring 2020 tend to be limited in their duration, while information-based voluntary social distancing continues beyond the end of lockdowns and likely encompasses a broader range of behaviors, such as wearing a mask, washing hands, or avoiding specific risky social situations, than could be effectively mandated by government

Beyond the characteristics of lockdown policies, it is likely that state fundamentals, such as population density, influence the effectiveness of voluntary social distancing. Fig 13 investigates this conjecture, by showing the correlations of voluntary social distancing effectiveness and population density, controlling for population. It shows that states with higher density

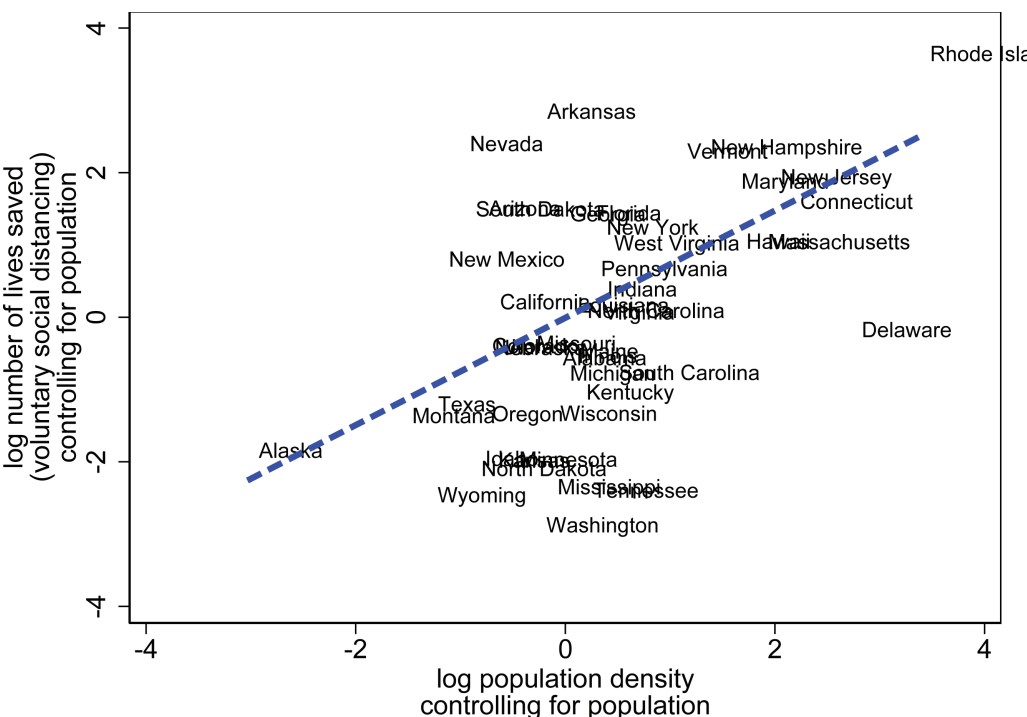

**Fig 13. Empirical relation between population density and number of lives saved by voluntary social distancing across US States**

tend to save more lives through information-based voluntary social distancing. This relationship is intuitive, as staying at home prevents people from spreading COVID-19 more in dense cities than rural areas. However, it should be noted that this relationship emerges from our model estimates despite the fact that we did not use any data on population density to estimate the model. It therefore serves as an additional "out-of-sample" prediction that confirms that our model produces generalizable regularities.

Fig 14 investigates the relationship between the average of the time-varying virality $R_t$ and the contagiousness of interactions $\psi$, following our discussion in Section 3.1.3. Interestingly, we find a non-monotonic relation between these two variables. For low values of $\psi$, increases in $\psi$ reduce average virality. However, past a value of 3, higher contagiousness of interactions $\psi$ is correlated with higher virality. This pattern makes sense if we consider average virality $R_t$ to be the balance of two opposing forces, as $R_t = \frac{\beta_0 \cdot m_t^\psi}{\gamma}$. In states with relatively low values of $\psi$, every reduction in mobility $m_t$ implies that infection rates can be more effectively reduced. However, as $\psi$ increases beyond 3, the effect that even little amounts of mobility $m_t$ can quickly spread the disease dominates. This explains why for very high values of $\psi$, virality is, on average, very high.

### 3.4 Externalities and the efficiency of social distancing

Our results can also be used to evaluate the relative economic costs of lockdowns as opposed to voluntary social distancing. To accomplish this, we calculate the mobility lost if lockdowns would have been used to save the same number of lives as voluntary social distancing. In other words, for each state, we calculate

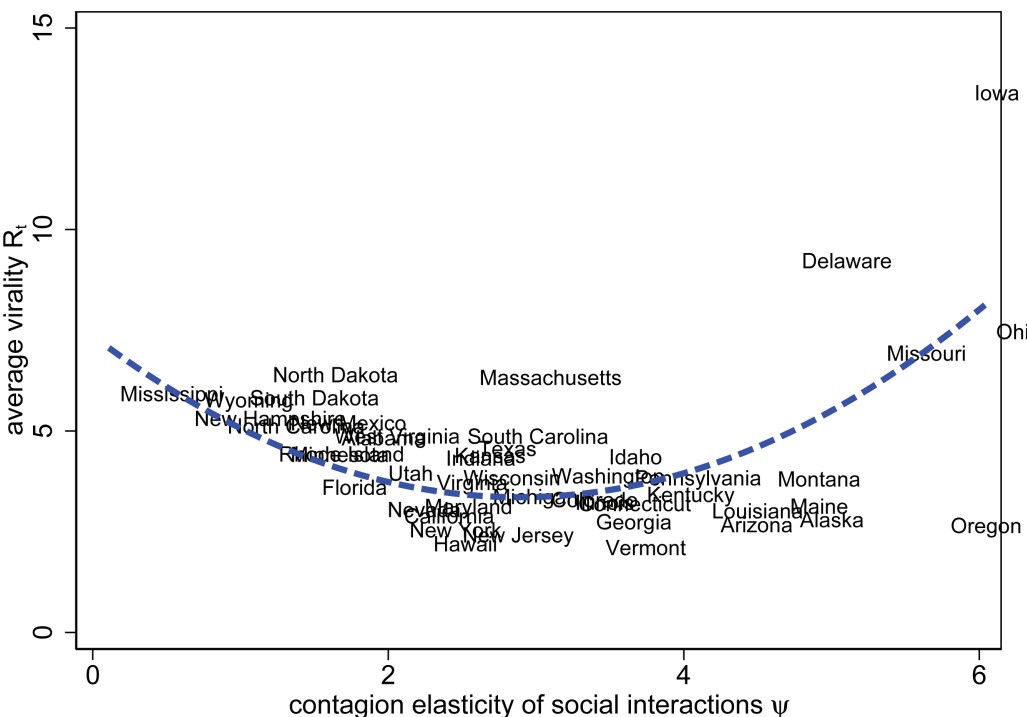

**Fig 14. Empirical relation between model-estimated contagion elasticity and average virality across US States.**

$$\xi = \frac{\text{lives saved by voluntary soc. dist.}}{\text{lives saved by lockdown}} \times \frac{\text{mobility lost due to lockdown}}{\text{mobility cost through voluntary soc. dist.}}, \qquad (36)$$

where $\xi$ measures how much more in terms of lost mobility it would have cost to save the same number of lives through lockdowns instead of voluntary social distancing. Our baseline results for $\xi$ across states are reported in the last column in Table 4. For the median state, the answer is –24.2%, suggesting that lockdowns would have avoided nearly a quarter of the economic costs associated with voluntary social distancing. However, it should be noted that there is a fair amount of variation in the efficiency of lockdowns. Indeed, as Table 4 shows, for some states, voluntary social distancing is far more efficient than lockdowns and implies that lockdowns would have cost 69.36% more in terms of lost mobility to save the same number of lives as voluntary social distancing has. As a consequence, lockdowns might be considered a targeted rather than general policy option over information policies. However, if blanket policies are the only option, then we find they are still economically beneficial on average.

A key factor influencing the relative efficiency of lockdowns as opposed to voluntary social distancing is the contagiousness of interactions $\psi$. Recall from equation (24) that $\psi$ governs the strength of negative health externalities from mobility. Under voluntary social distancing, stronger externalities implied by higher $\psi$ lead to more exposure and, ultimately, more infections in the future. More infections, in turn, depress mobility via equation (28). As a result, higher contagiousness of interactions leads to more disease spread under voluntary social distancing, leading to stronger social distancing. In contrast, imposing lockdowns reduces infections in the first place, so that subsequent mobility can be higher as the number of confirmed

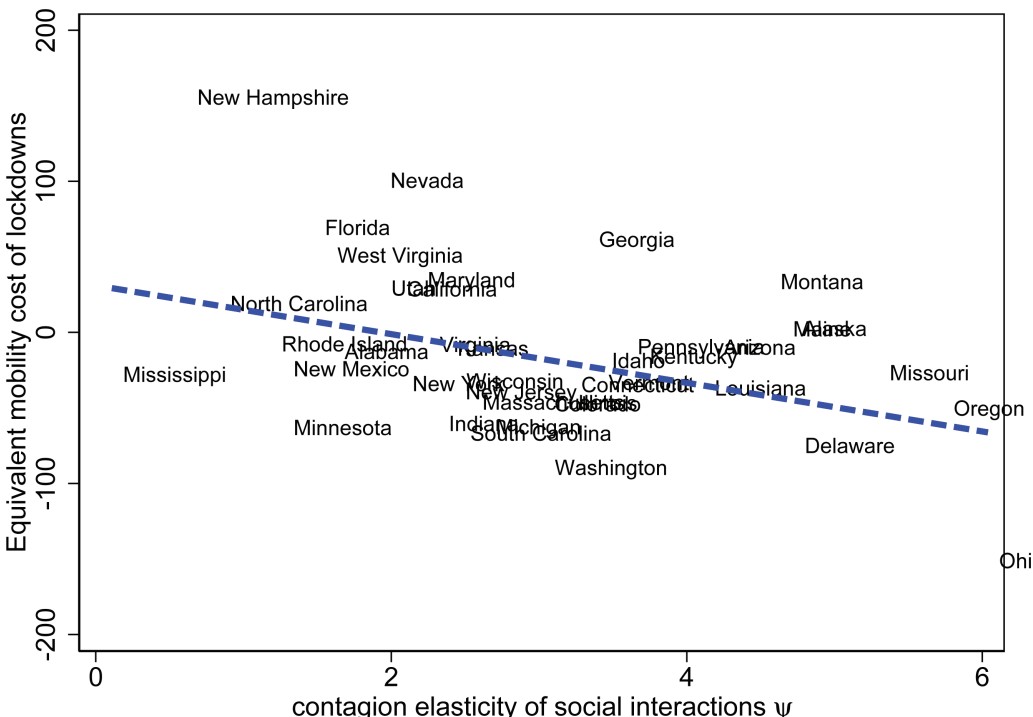

**Fig 15. Empirical relation between model-estimated mobility costs of lockdowns and model-estimated contagion elasticity across US States.**

cases is lower. Evidence for this mechanism is presented in Fig 15. The y-axis shows measures of $\xi$ in percentage points, with lower values capturing higher efficiency of state lockdowns. The x-axis captures the contagiousness of the interaction parameter $\psi$. Fig 15 shows that states with higher estimates for $\psi$ and therefore stronger health externalities exhibit higher relative efficiency of state lockdowns compared to voluntary social distancing.

### 3.5 Information policy counterfactuals

The results in the previous section raise the question of how much changes in information policy—which influence the parameters $\mu_0$, $\mu_1$, and $\mu_2$—matter for saving lives during COVID-19. This approach for using changes in $\mu_0$, $\mu_1$, and $\mu_2$ as proxies for changes in information policies follows from our theoretical discussion in section 3.2.6, where we showed that changes in information quality $\rho_\epsilon$ lead to changes in $\mu_0, \mu_1, \mu_2$. To discipline this quantitative exercise, we return to the two states of West Virginia and Massachusetts. Our estimates of coefficients $\mu_0$, $\mu_1$, and $\mu_2$ are consistent with the view that information quality in West Virginia is poor, while people have a higher prior on the base risk of infection. In contrast, the $\mu_0$, $\mu_1$, and $\mu_2$ estimates for Massachusetts suggest that people believe information quality to be high, while their priors about base infection rates are low.

In order to contrast differences in information quality, expectation formation, and voluntary social distancing, we impose either West Virginia's or Massachusetts' mobility coefficients $\mu_0$, $\mu_1$, and $\mu_2$ on all states. Then we recalculate the effectiveness of social distancing by taking the difference between lives saved by voluntary social distancing with the alternative uniform parameters $\mu_0$, $\mu_1$, and $\mu_2$ as opposed to our baseline estimates with heterogeneous information quality.

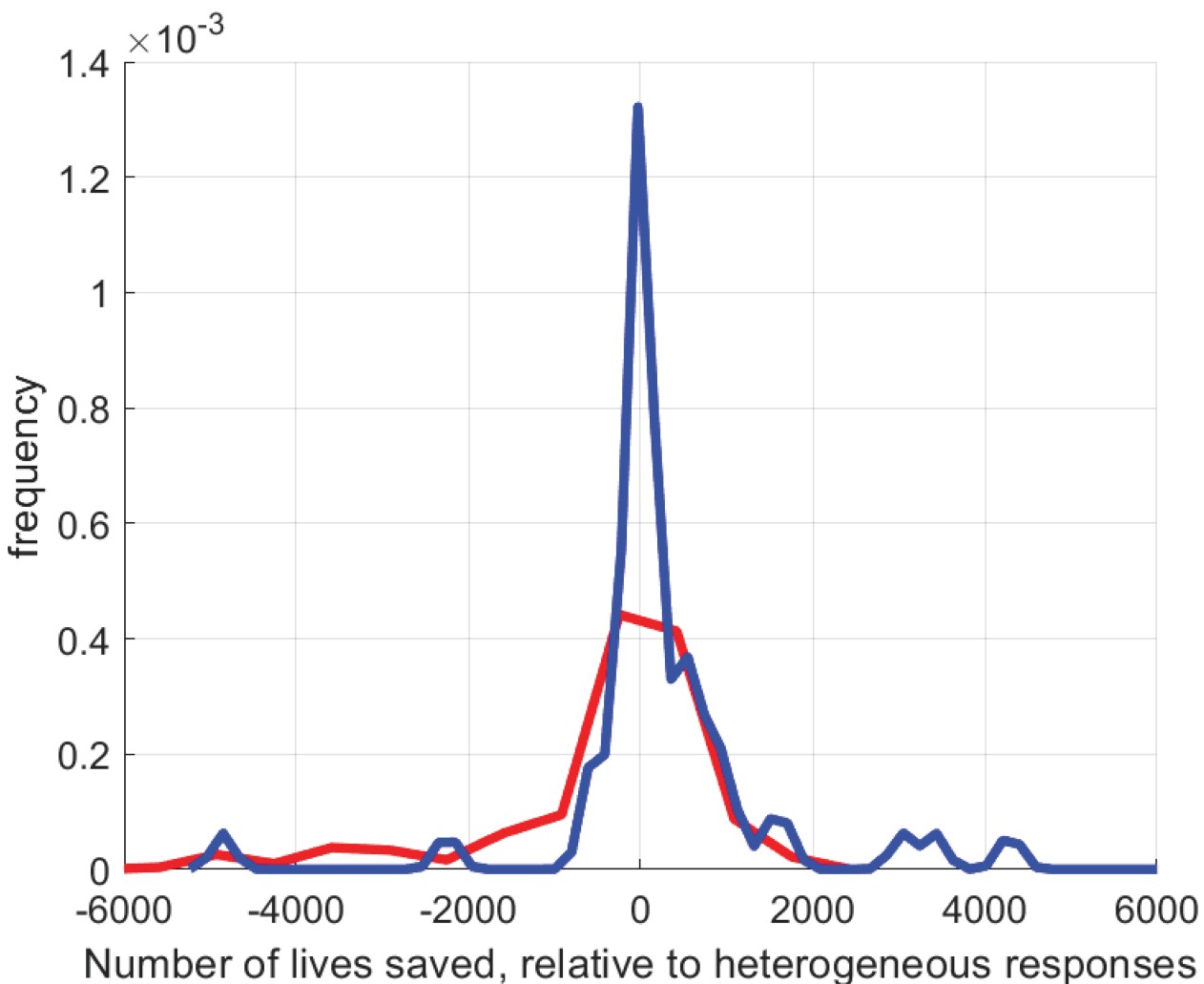

**Fig 16. Distribution of heterogeneous responses to counterfactual voluntary social distancing behavior.**

We report our results in Table 5. The entries capture the sum of lives saved across all states. As the first entry shows, more than 246,000 additional people would have died if people in all states followed the same expectation formation process as people in West Virginia. This number is a substantial counterfactual increase in fatalities, compared to around 100,000 deaths by the end of June 2020. In contrast, 26,071 more lives would have been saved if everyone trusted published case counts as much as people in Massachusetts. Imposing uniform mobility responses underlines the role played by different coefficients. Recall from Fig 8 that Massachusetts had a very high value for $\mu_0$ but also very high absolute values for $\mu_1$ and $\mu_2$. While the higher value of $\mu_1$ tends to increase the number of lives saved across states, the higher value of $\mu_0$ tends to reduce it. Therefore, the worst combination is Massachusetts's high unconditional mobility $\mu_0$, which might reflect more optimism about the base infection risk with West Virginia's weak responsiveness to published case and fatality counts $\mu_1$ and $\mu_2$. This combination would have implied an additional 1.6 million fatalities. In contrast, the best combination of $\mu_0$, $\mu_1$, and $\mu_2$ would have saved an additional 116,589 people.

**Table 5. Lives saved by uniform mobility responses for different mobility parameters.**

| | | $\mu_1, \mu_2$ | |
| --- | --- | --- | --- |
| | | WV | MA |
| $\mu_0$ | WV | −246,635 | 116,589 |
| | MA | −1,664,915 | 26,071 |

*Note:* Number of lives saved relative to estimates mobility parameters $\mu_0, \mu_1, \mu_2$, with negative numbers indicating higher fatalities. Entries are calculated are predicted fatalities under uniform mobility parameters, based on responses in West Virginia (WV) or Massachusetts (MA) minus fatalities under estimated current responses. Entries are cumulative until end of June 2020.

An important finding of Table 5 is that there exists significant asymmetry in the importance of bad and good information policies: bad information policies harm much more than good information policies help. Fig 16 illustrates what drives this asymmetry. Specifically, it displays the distribution of lives saved for the uniform West Virginia expectations in red, while the uniform Massachusetts expectations are in blue. This figure shows that the additional lives lost due to low-quality information are much more concentrated at very low values. In other words, low-quality information disproportionately harms states that already have a bad outbreak. In contrast, the effects of high-quality information are much more heterogeneous, harming some states while helping other states.

## 4 Discussion and contribution to the literature

We develop a model of information-based voluntary social distancing that is rich enough to quantify the effect of credible local information on case and fatality counts. Our model incorporates the insights highlighted by empirical studies that document, but do not model, the impact of beliefs and the information environment on social distancing ([17], [30], [31]). Our approach extends work by [8], who model social distancing with a compartmental model, by allowing social distancing to be endogenous to the information environment. Our extension is critical to understand what policies or other determinants can influence voluntary social distancing.

This paper also contributes to a small literature of hybrid approaches between Machine Learning and traditional compartmental epidemiological models (henceforth "hybrid approach"). Some hybrid approaches such as [32] and [33] use very basic compartmental models, which do not allow for voluntary social distancing at all. Based on the literature review of 136 COVID-19 forecasting models in [1], we identified 3 studies that use both a hybrid approach and daily mobility data: [34], [35] and [36]. Our paper extends these papers by also allowing for a feedback from cases and fatalities to mobility through an information channel. To the best of our knowledge, ours is therefore the only forecasting and policy model that utilizes a hybrid approach while allowing for a feedback of local information on cases and fatalities to mobility.

Another hybrid approach we are aware of is pursued by [37], who combine an epidemiology-founded version of Bayes' Law with Machine Learning-based infection-prediction using symptoms to provide a sample-selection corrected measure of the active prevalence of COVID-19. However, their approach does not allow for counterfactual analysis of NPIs, such as lockdown policies.

We believe that our combination of a hybrid model with an information-based theory of voluntary social distancing has three unique advantages for policy makers seeking to manage the early stages of future pandemics. First, the incorporation of Machine Learning avoids over-fitting and provides more reliable short-run predictions than either purely statistical

models or traditional compartmental models with unmitigated exponential growth. Second, our framework provides realistic estimates of the causal effects of lockdowns, taking into account that even without lockdowns people would voluntarily socially distance. Crucially, the extent of voluntary social distancing and therefore the causal estimate of lockdowns depend in our model on local parameters including the credibility of locally reported data. Any policy analysis ignoring how voluntary social distancing differs locally is likely to provide biased estimates of the economic and health effects of lockdown policies. Third, our framework allows for a credible cost-benefit analysis of the trade-off between lives and livelihoods when deploying lockdown policies in early phases of a pandemic.

## 5 Conclusion

This paper has developed a new methodology to evaluate the effectiveness of social distancing during COVID-19. To achieve this, we combine an extended compartmental model of a pandemic with Machine Learning and a new theory of information-based voluntary social distancing.

Qualitatively, the model shows that information-based voluntary social distancing was an important feature of the COVID-19 policy response. Quantitatively, our calibrated model suggests the size of this effect may be large enough to be important. Of course, the exact numbers depend on specific calibrations and other model assumptions. The model we provide is tractable enough for future researchers to use and update with different calibrations or model assumptions.

Future research can use our model to analyze at least three additional policy-relevant questions. First, how do different policy alternatives, such as (1) proactive testing and quarantining, (2) increased symptom-based testing, and (3) efforts to increase public attention to published case counts, quantitatively differ in slowing the spread of pandemics like COVID-19? Second, what are the economic implications of these different non-pharmaceutical interventions? Our framework already quantified the reduction in mobility-based economic activities, such as going to work and grocery shopping, but we have not directly translated these mobility changes into unemployment or GDP numbers (although doing so, using regularities such as Okun's Law is straightforward). Third, are there important policy complementarities between different non-pharmaceutical interventions? For example, more aggressive testing might increase case counts, while governments can also influence the degree of voluntary social distancing by focusing the public's attention. Might a combination of these two policies disproportionately slow the spread of the virus down? These questions can be addressed by the framework developed in this paper, and we leave them for future research.

## Notes

[1] A full review of the entire literature on COVID-19 forecasting models is beyond the scope of this paper, but see [1] for a literature review including 136 papers.

[2] A related but distinct learning mechanism on NPIs is modeled by [9], who show that local mask mandates can affect voluntary social distancing through changes in perceived local infection risks.

[3] see: https://www.google.com/covid19/mobility/

[4] See, for example, the mobility to parks in the Google global mobility report.

[5] See "Trump Says Coronavirus Cure Cannot 'Be Worse Than the Problem Itself'," NY Times, March 23, 2020.

[6] Note that while we do not control directly for county or city level lockdown measures, their timing will be accounted for, if not their severity, if they are concomitant with state-wide lockdowns.

[7] It should be noted that we assume that people are equally infectious, given their behavior. In other words, in our model differences in infectiousness across people is the result of differences in type of behavior pursued by these people. From this perspective, "super-spreader" individuals infect more people because they attend super-spreader events or do not socially distance much and not because of fixed traits. It would be relatively straightforward to generalize the matching function of individuals to model such differences in fixed individual infectiousness.

[8] The model could easily be extended to incorporate differences in death rates of detected and undetected cases.

[9]It is worth noting that the optimal mobility equation (28) naturally generates mean-reversion in voluntary social distancing over time, given the disease dynamics of our model. The reason for this mean reversion is that in the beginning stages of an infection, the number of confirmed cases will strongly increase, while not many people will have died from COVID-19. Therefore, early on, voluntary social distancing will depress mobility significantly. However, as the number of fatalities grows, people update their perceived infection risk downward, according to equation (26). This effect partially offsets increased social distancing from growing confirmed case numbers.

[10]Results from these simulations are available upon request.

[11]The summary statistics in this section are mostly based on statistics from the US Census Bureau for 2019.

[12]The $95^{th}$ percentile of the state lockdown effects are 20% mobility reduction each day.

## Author contributions

**Conceptualization:** Mu-Jeung Yang.

**Data curation:** Mu-Jeung Yang.

**Formal analysis:** Mu-Jeung Yang.

**Investigation:** Mu-Jeung Yang, Nathan Seegert.

**Methodology:** Mu-Jeung Yang.

**Software:** Yang Fan.

**Validation:** Yang Fan.

**Writing – original draft:** Mu-Jeung Yang.

**Writing – review & editing:** Maclean Gaulin, Nathan Seegert, Yang Fan.

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
