## [Editor Report · Decision Letter 0]

4 Nov 2022

PONE-D-22-26653What Drives the Effectiveness of Social Distancing in Combating COVID-19 across U.S. States?PLOS ONE

Dear Dr. Yang,

Thank you for submitting your manuscript to PLOS ONE. After careful consideration, we feel that it has merit but does not fully meet PLOS ONE’s publication criteria as it currently stands. Therefore, we invite you to submit a revised version of the manuscript that addresses the points raised during the review process.

The manuscript explores an important aspect of the pandemic- lockdown and economic implications using machine learning algorithms and various modeling.  Though interesting the styling/formatting of the manuscript is not in accordance with PLoSOne's guidelines. We prefer Vancouver style and not AMA style or using foot notes throughout the text.  Also, a more economics focused journal for submission may be considered (suggestion only). Given the niche nature of the field, delays with reviewer search may be taken into account with revised submission. 

We look forward to receiving your revised manuscript.

Kind regards,

Vineet Gupta, MD, FACP, SFHM

Academic Editor

PLOS ONE

Journal Requirements:

3. Please upload a new copy of Figures 13, 14 and 16 as the detail is not clear. Please follow the link for more information:

https://blogs.plos.org/plos/2019/06/looking-good-tips-for-creating-your-plos-figures-graphics/

https://blogs.plos.org/plos/2019/06/looking-good-tips-for-creating-your-plos-figures-graphics/

---

## [Author Response · Author response to Decision Letter 1]

25 Nov 2022

No reviews in the first round. We changed the bibliography formatting to Vancouver.

---

## [Editor Report · Decision Letter 1]

31 Jul 2023

PONE-D-22-26653R1What Drives the Effectiveness of Social Distancing in Combating COVID-19 across U.S. States?PLOS ONE

Dear Dr. Yang,

Thank you for submitting your manuscript to PLOS ONE. After careful consideration, we feel that it has merit but does not fully meet PLOS ONE’s publication criteria as it currently stands. Therefore, we invite you to submit a revised version of the manuscript that addresses the points raised during the review process.

We look forward to receiving your revised manuscript.

Kind regards,

Alireza Bornamanesh

Guest Editor

PLOS ONE

Additional Editor Comments;

I enjoyed reading this manuscript, and believe that it holds promise. At the same time, I identified several interconnected issues that require the authors’ attention before the manuscript can be considered for publication. First, the main argument needs to be worked out and formulated much more clearly. The main argument should be included in the paper’s introduction. Second, a clear argument will make it easier for the authors to highlight their main contribution to the existing academic literature, which is relatively vague right now. Furthermore, the discussion and conclusions are difficult to follow. I think that they need to be rewritten to better reflect the 

---

## [Author Response · Author response to Decision Letter 2]

15 Jan 2024

Dear Editor,

Thank you very much for the opportunity to revise our work, for potential publication in PLOS One. We have followed your advice closely and have reworked the introduction and discussion section of the paper to more clearly highlight the contribution of our work to the literature.

We believe that this has helped to strengthen the paper considerably, and are indebted to you for your guidance. Please see below a detailed response to your previous review. We have reprinted your comments in bold and are responding in normal font.

I enjoyed reading this manuscript, and believe that it holds promise. At the same time, I identified several interconnected issues that require the authors’ attention before the manuscript can be considered for publication. First, the main argument needs to be worked out and formulated much more clearly. The main argument should be included in the paper’s introduction.

Thank you for this excellent suggestion. We now highlight our main contribution already in the abstract, which we reprint here for your convenience:

“We propose a new theory of information-based voluntary social distancing in which people's responses to disease prevalence depend on the credibility of reported cases and fatalities and vary locally. We embed this theory into a new pandemic prediction and policy analysis framework that blends compartmental epidemiological/economic models with Machine Learning. We find that lockdown effectiveness varies widely across US States during the early phases of the COVID-19 pandemic. We find there would have been 240,000 more fatalities by June 2020 if people in all states would have voluntarily socially distanced like people in the least informed US state, and 25,000 fewer fatalities if people in all states would have responded like people in the most informed state.”

Additionally, we extensively discuss our main contribution in the introduction (see below).

Second, a clear argument will make it easier for the authors to highlight their main contribution to the existing academic literature, which is relatively vague right now.

Thank you for this suggestion. We have followed your advice and completely rewritten the introduction of the paper to more clearly emphasize the contribution to the academic literature. We reprint the key passages discussing our contribution to the literature in the following:

“This paper complements the existing literature through three distinct contributions. Our first contribution is a new theoretical model of information-based voluntary social distancing, in which people learn about infection risks from publicly available data on new cases and fatalities. Importantly, the strength of voluntary social distancing responses to reported local cases and fatalities is driven by the perceived credibility of this data, which can be influenced by local governments' information policies and which will vary locally. In this context, our model predicts that local information policies that produce less credible case and fatality counts lead to a lower signal-to-noise ratio of published case and fatality counts. As a consequence, local information policies will impact the degree of voluntary social distancing directly and the effectiveness of lockdown policies indirectly. This insight goes well beyond the analysis of [7] and has important policy implications. We are not aware of any other paper in the literature providing this insight or any similar policy analysis.

Our second contribution is methodological, in that we propose a new prediction and policy analysis framework combining an extended compartmental epidemiological model with Machine Learning. We embed our theory of information-based social distancing into a rich compartmental model including a variety of often unobserved, time-varying factors such as asymptomatic transmission, symptom-based testing and quarantining, and time-variation in fatality rates. In addition to daily local data on new COVID-19 cases and fatalities, we calibrate the model to local, daily cellphone-based data on mobility, which we establish as valid proxy for economic activity. Our framework can therefore provide cost-benefit analysis for policy makers interested in the economic and health effects lockdown policies. To increase the reliability of model forecasts, we use ensemble learning and cross-validation, two ideas from Machine Learning. Specifically, we re-calibrate the model for different time horizons and then use a weighted average of these estimates, to increase the robustness of out-of-sample predictions. The ensemble weights are then chosen via cross-validation, which minimizes the out-of-sample prediction error. Using these additional steps allows us to generate more generalizable patterns that are less likely driven by statistically noisy initial conditions. Of the 136 COVID-19 forecasting models reviewed in [1], only 3 papers use a hybrid approach between Machine Learning and compartmental models in combination with mobility data. All of these papers use mobility data as a predictor for new cases and fatalities but none of these three papers allows for a feedback of case and fatality data to mobility, the way our paper does. This feedback is a direct consequence of our theory of information-based voluntary social distancing and is therefore a main focus of our analysis.

Our third contribution consists of novel empirical results from the application of our new model and methodology to the 50 US states during the early phase of COVID-19 in 2020. We find that local differences in mobility responses and other parameters imply a wide variation of the economic efficiency of lockdown policies across states. To quantify economic efficiency of lockdown policy (henceforth ``lockdown efficiency") we use mobility as a proxy for economic activity and measure mobility lost if lockdowns would have been used to save the same number of lives as have been saved through information-based voluntary social distancing in a given state. We find that lockdown efficiency varies widely. For example, for US states in the 25th percentile of the lockdown efficiency distribution, lockdowns could have saved the same number of lives as voluntary social distancing, but allowing for 75% more mobility until June 2020. In contrast, for US states in the 75th percentile of the lockdown efficiency distribution, using lockdowns instead of voluntary social distancing would have implied almost 20% less economic activity. In other words, the efficiency of lockdown policies varies substantially across states. To investigate the role of differences in local information-based voluntary social distancing, we impose different mobility responses to reported case and fatality counts. Our information-based voluntary social distancing model suggests that these mobility responses can be systematically influenced by the credibility of local government reports. To provide a reasonable counterfactual for changes in credibility of local government reports, we compare the number of lives saved by uniformly imposing voluntary social distancing responses in places with low information transmission like West Virginia compared to relatively high information transmission places like Massachusetts. Using West Virginia parameters for the whole US implies over 240,000 additional fatalities before June 2020. In contrast, imposing parameters from Massachusetts across all US states only saves an additional 24,000 lives before June 2020. This shows that the existence of important cost-benefit asymmetries of these information policies, as low-credibility information policies are much more costly in terms of fatalities than high-credibility information policies are beneficial in saving lives. Our quantitative evidence on how local information policies effect pandemic outcomes, to our knowledge unique to this study, highlights the importance of credible information due to voluntary social distancing.“

Furthermore, the discussion and conclusions are difficult to follow. I think that they need to be rewritten to better reflect the interesting research results and to highlight the authors’ key contributions.

Thank you for pointing this out. In response, we have added are more detailed discussion, highlighting our key contributions. We reprint the key paragraphs here for your convenience:

“We develop a model of information-based voluntary social distancing that is rich enough to quantify the effect of credible local information on case and fatality counts. Our model incorporates the insights highlighted by empirical studies that document, but do not model, the impact of beliefs and the information environment on social distancing ([16], [29], [30]). Our approach extends work by [7], who model social distancing with a compartmental model, by allowing social distancing to be endogenous to the information environment. Our extension is critical to understand what policies or other determinants can influence voluntary social distancing.

This paper also contributes to a small literature of hybrid approaches between Machine Learning and traditional compartmental epidemiological models (henceforth ``hybrid approach"). Some hybrid approaches such as [31] and [32] use very basic compartmental models, which do not allow for voluntary social distancing at all. Based on the literature review of 136 COVID-19 forecasting models in [1], we identified 3 studies that use both a hybrid approach and daily mobility data: [33], [34] and [35]. Our paper extends these papers by also allowing for a feedback from cases and fatalities to mobility through an information channel. To the best of our knowledge, ours is therefore the only forecasting and policy model that utilizes a hybrid approach while allowing for a feedback of local information on cases and fatalities to mobility.

Another hybrid approach we are aware of is pursued by [36], who combine an epidemiology-founded version of Bayes' Law with Machine Learning-based infection-prediction using symptoms to provide a sample-selection corrected measure of the active prevalence of COVID-19. However, their approach does not allow for counterfactual analysis of NPIs, such as lockdown policies.”

Thank you again for your guidance and constructive feedback. Your suggestions have helped to considerably strengthen the paper.

Sincerely, The authors

---

## [Decision Letter · Decision Letter 2]

15 May 2024

PONE-D-22-26653R2What Drives the Effectiveness of Social Distancing in Combating COVID-19 across U.S. States?PLOS ONE

Dear Dr. Yang,

Thank you for submitting your manuscript to PLOS ONE. After careful consideration, we feel that it has merit but does not fully meet PLOS ONE’s publication criteria as it currently stands. Therefore, we invite you to submit a revised version of the manuscript that addresses the points raised during the review process. We considered necessary inviting additional reviewers to assess this work. We have now received 4 completed reviews; the comments are available below. Most reviewers are happy with your manuscript but Reviewer#4 has raised significant scientific concerns about the study that need to be addressed in a revision.

We look forward to receiving your revised manuscript.

Kind regards,

Miquel Vall-llosera Camps

Senior Staff Editor

PLOS ONE

Reviewers' comments:

Reviewer's Responses to Questions

**Comments to the Author**

1. If the authors have adequately addressed your comments raised in a previous round of review and you feel that this manuscript is now acceptable for publication, you may indicate that here to bypass the “Comments to the Author” section, enter your conflict of interest statement in the “Confidential to Editor” section, and submit your "Accept" recommendation.

Reviewer #1: All comments have been addressed

Reviewer #2: All comments have been addressed

Reviewer #3: All comments have been addressed

Reviewer #4: (No Response)

2. Is the manuscript technically sound, and do the data support the conclusions?

Reviewer #1: Yes

Reviewer #2: Yes

Reviewer #3: Yes

Reviewer #4: Partly

3. Has the statistical analysis been performed appropriately and rigorously? 

Reviewer #1: Yes

Reviewer #2: Yes

Reviewer #3: Yes

Reviewer #4: Yes

4. Have the authors made all data underlying the findings in their manuscript fully available?

Reviewer #1: Yes

Reviewer #2: Yes

Reviewer #3: Yes

Reviewer #4: Yes

5. Is the manuscript presented in an intelligible fashion and written in standard English?

Reviewer #1: Yes

Reviewer #2: Yes

Reviewer #3: Yes

Reviewer #4: Yes

6. Review Comments to the Author

Reviewer #1: Paper is well written and can be accepted after the following corrections:(1) There is overlap in text in Figures 2, 5, 6, .... Authors are suggested to update all these figures and ensure their proper representation.(2) Caption of Figure 15 is not informative as it does not explain blue and red curves.(3) Authors are suggested to update the caption of Figure to make them more informative.(4) Reproducibility of the study is important for reliability of the work and also for future research and improvements. Therefore, authors are suggested to provide computer code/programmes developed and used for this study in some publicly open repository like GitHub/ResearchGate or some other open source storage.(5) Authors are suggested to provide limitations and future directions of the research.

Reviewer #2: Try to improve Figure No. 13, 14, and 16. It is not very clear. And add some good reference in the literature, i. e. 1. Singh, A., & Bajpai, M. K. (2020). SEIHCRD model for COVID-19 spread scenarios, disease predictions and estimates the basic reproduction number, case fatality rate, hospital, and ICU beds requirement. Computer Modeling in Engineering & Sciences, 125(3), 991-1031.

2. Singh, A., Bajpai, M. K., & Gupta, S. L. (2023). A Time-dependent mathematical model for COVID-19 transmission dynamics and analysis of critical and hospitalized cases with bed requirements. In Machine Vision and Augmented Intelligence: Select Proceedings of MAI 2022 (pp. 85-120). Singapore: Springer Nature Singapore.

3. Singh, Avaneesh, Saroj Kumar Chandra, and Manish Kumar Bajpai. "Study of non-pharmacological interventions on COVID-19 spread." Computer Modeling in Engineering & Sciences 125, no. 3 (2020): 966-989.

4. Singh, Avaneesh, and Manish Kumar Bajpai. "A compartmental Mathematical model of COVID-19 intervention scenarios for Mumbai." In Machine Vision and Augmented Intelligence: Select Proceedings of MAI 2022, pp. 121-146. Singapore: Springer Nature Singapore, 2023.

Reviewer #3: The article is relevant, very well written and with important results. The authors made the corrections proposed by the other reviewers.

Reviewer #4: Thanks for giving me the opportunity to review this manuscript. The retrospective investigation of the effectiveness of social distancing is an important step in preparing for future pandemics. The manuscript introduces various thoughts and deliberation that shed new light on this complex system.

On the other hand, I am afraid that the presented model outcomes are highly uncertain as the model involves a large number of calibration parameters. Moreover, some of the parameter estimates, which are presented as model results, seem to be rather unrealistic. For instance, authors claim that death rates have been fallen within 27 days from 10% to 0.003% due to changes in treatment alone. The infection fatality rate is estimated to 9.15% at the beginning of the outbreak. According to the model this value eventually dropped to 0.003%. Similarly, the basic reproduction seems to be too high (R0=6). Results of other studies significantly deviate from the numbers presented in this study.

Moreover, R0 and IFR are highly dependent on the local geography of social networks, culture, habits, age structures, health care, seasonality and many other factors. In addition to uncertainties in parameter estimates, misperceptions may also originate from the modeling paradigm being used. For instance, the aggregated (equation based) and distributed models (individual or agent based) tend to produce very different results even if the input parameters are the same.

In the light of these arguments, I’m in doubt that a model of this type can be used to make quantitative forecasts or retrospective estimates on fatality numbers under given conditions. To deal with that problem I suggest to

1) Put a focus on qualitative model outcomes (e.g. voluntary social distancing is more effective than state lockdowns, regional differences and reason why, economic implications etc.). These are the results that should be highlighted in the abstract (e.g. …results show that effect of voluntary social distancing is more important than the impact of state lockdowns…).

2) If quantitative results are presented, compare with results of other studies (IFR, R0 etc.).

3) Avoid mentioning quantitative results that cannot be validated (e.g. fatalities under hypothetical conditions) or at least add a statement on the uncertainty associated with such estimates.

4) Discuss the limitations and uncertainties of calibration procedures. In theory, calibrated model parameters can be wrong while model outcomes (like fatalities and cases over time) correctly reproduce statistical records.

5) Where applicable check your result for plausibility. For instance, other studies showed that the proportion of asymptomatic is dependent on age structures. Are your results consistent with that assumption?

6) Address limitations of the data: For instance, Google mobility data is used to quantitatively measure people’s response: Wearing face masks or keeping a distance may lead to a decoupling between mobility volumes and infectious contacts. Mobility data alone cannot address such dependencies.

Some remarks concerning the formal appearance:

.) Results are presented in the introduction section (lines 80-110), which is kind of unusual. I am however not quite familiar with the formal criteria that are used by PLOS.

.) Figure 2 show changes in mobility and unemployment. Mobility reduction is associated with higher unemployment. Does the change in unemployment happen between April 2019 and April 2020? What is meant by year-over-year? Please specify in figure 2. It is also not clear how you use Google Mobility data in Fig. 2. Did you take the average of economically relevant categories such as mobility for work, grocery shopping etc.? Moreover, mobility reduction may have caused unemployment. Alternatively, unemployment may have caused a reduction in workplace mobility. Please explain.

.) In line 167 you say that Fig. 3 shows relative mobility in June. According to the y-axis of Fig. 3 it is spring 2020. What month or season is shown on the y-axis?

.) Fig.4 : What time intervals and time span are used? Spring in weekly time steps? Figures are not self-explanatory!

.) The correlations presented in Fig. 6 and 7 seem to be rather weak. Please provide coefficients.

.) In line 202 it is stated that Fig 8 shows negative correlation. This looks like positive correlation.

.) I suggest including full description of abbreviations in Fig. 10, which would significantly improve readability.

.) line 310: In other words….

I am looking forward to seeing your revised manuscript!

7. PLOS authors have the option to publish the peer review history of their article (what does this mean?). If published, this will include your full peer review and any attached files.

Reviewer #1: No

Reviewer #2: **Yes: **Avaneesh Singh

Reviewer #3: No

Reviewer #4: No

---

## [Author Response · Author response to Decision Letter 3]

3 Jul 2024

Please see the response letters uploaded under "Responses to reviewers"

---

## [Decision Letter · Decision Letter 3]

22 Jul 2024

What Drives the Effectiveness of Social Distancing in Combating COVID-19 across U.S. States?

PONE-D-22-26653R3

Dear Dr. Yang,

We’re pleased to inform you that your manuscript has been judged scientifically suitable for publication and will be formally accepted for publication once it meets all outstanding technical requirements.

Kind regards,

Mike Farjam

Academic Editor

PLOS ONE

Additional Editor Comments (optional):

The paper has been under review for years now, and 3 out of 4 reviewers say it can be accepted as is, so I accept it. Reviewer 4 has some minor points that I invite you to look at, and I leave it up to you to decide whether you want to make some very minor changes to address them for the final version that gets published.

Reviewers' comments:

Reviewer's Responses to Questions

**Comments to the Author**

1. If the authors have adequately addressed your comments raised in a previous round of review and you feel that this manuscript is now acceptable for publication, you may indicate that here to bypass the “Comments to the Author” section, enter your conflict of interest statement in the “Confidential to Editor” section, and submit your "Accept" recommendation.

Reviewer #1: All comments have been addressed

Reviewer #2: All comments have been addressed

Reviewer #3: All comments have been addressed

Reviewer #4: All comments have been addressed

2. Is the manuscript technically sound, and do the data support the conclusions?

Reviewer #1: Yes

Reviewer #2: Yes

Reviewer #3: Yes

Reviewer #4: Yes

3. Has the statistical analysis been performed appropriately and rigorously? 

Reviewer #1: Yes

Reviewer #2: Yes

Reviewer #3: Yes

Reviewer #4: Yes

4. Have the authors made all data underlying the findings in their manuscript fully available?

Reviewer #1: No

Reviewer #2: Yes

Reviewer #3: Yes

Reviewer #4: (No Response)

5. Is the manuscript presented in an intelligible fashion and written in standard English?

Reviewer #1: Yes

Reviewer #2: Yes

Reviewer #3: Yes

Reviewer #4: Yes

6. Review Comments to the Author

Reviewer #1: Authors agreed to the following comments, so they should submit the code to GitHub and provide the link of the same in the final version of the paper "Reproducibility of the study is important for reliability of the work and also for future research and improvements. Therefore, authors are suggested to provide computer

code/programmes developed and used for this study in some publicly open repository like

GitHub/ResearchGate or some other open source storage."

Reviewer #2: (No Response)

Reviewer #3: The authors made all the previously corrections proposed by me and by the other reviewers. Paper is well written and can be accepted with no more considerations.

Reviewer #4: You write…

“First, we impose an upper bound of death rates for symptomatic people of 15%, consistent with the case fatality rate of 15%, which prevailed in Italy at the height of the COVID-19 crisis in that country. Italy’s case fatality ratio, in turn, is the highest currently reported case fatality ratio in the world.“

This implies that Italy managed to detect all the symptomatic cases, which is very unlikely. Indeed, this number is a suitable upper bound value given that in reality the fatality ratio of symptomatic is probably lower.

However, according to WHO-Data the Italian CFR never exceeded 5%. Where did you get this number? The discrepancy could be due to the averaging of CFR though.

---

I am surprised about the nice relationship between mobility and confirmed cases in Fig. 4. You say that “every point is a different day in April 2020”. Shouldn’t there be a time lag between those variables? I’d assume a time lag of about 5 days. Does the correlation increase when adding a time lag? In theory, less mobility could be caused by high prevalence of the disease. Such a reverse causality could be detected by experimenting with time lags.

---

Overall, you could significantly improve the manuscript. Your research nicely illustrates an idea. The empirical prove is constrained by data quality and the complexity of the system as such. I guess many systems scientists are confronted with that challenges that make a retrospective evaluation of interventions and strategies quite demanding.

7. PLOS authors have the option to publish the peer review history of their article (what does this mean?). If published, this will include your full peer review and any attached files.

Reviewer #1: **Yes: **Pushpendra Singh, PhD, School of Engineering, JNU Delhi, India

Reviewer #2: No

Reviewer #3: No

Reviewer #4: No

---

## [Editor Report · Acceptance letter]

PONE-D-22-26653R3

PLOS ONE

Dear Dr. Yang,

I'm pleased to inform you that your manuscript has been deemed suitable for publication in PLOS ONE. Congratulations! Your manuscript is now being handed over to our production team.

Kind regards,

on behalf of

Dr. Mike Farjam

Academic Editor

PLOS ONE